# META-LEARNING THEORY-INFORMED INDUCTIVE BIASES USING DEEP KERNEL GAUSSIAN PROCESSES

**Bahti Zakirov**
Institute of Science and Technology Austria
bzakirov@ist.ac.at

**Gašper Tkačik**
Institute of Science and Technology Austria
gtkacik@ist.ac.at

## ABSTRACT

Normative and task-driven theories offer powerful top-down explanations for biological systems, yet the goals of quantitatively arbitrating between competing theories, and utilizing them as inductive biases to improve data-driven fits of real biological datasets are prohibitively laborious, and often impossible. To this end, we introduce a Bayesian meta-learning framework designed to automatically convert raw functional predictions from normative theories into tractable probabilistic models. We employ adaptive deep kernel Gaussian processes, meta-learning a kernel on synthetic data generated from a normative theory. This *Theory-Informed Kernel* specifies a probabilistic model representing the predictions of a given theoretical model: usable for both fitting data and rigorously validating the theory. As a demonstration, we apply our framework to the early visual system, using efficient coding as our normative theory. We show improved response prediction accuracy in *ex vivo* recordings of mouse retinal ganglion cells stimulated by natural scenes compared to conventional data-driven baselines, while providing accurate uncertainty estimates and interpretable representations. Using exact Bayesian model selection, we also show that our informed kernel can accurately infer the *degree* of theory-match from data, confirming faithful encapsulation of theory structure. This work provides a more general, scalable, and automated approach for integrating theoretical knowledge into data-driven scientific inquiry in neuroscience and beyond.

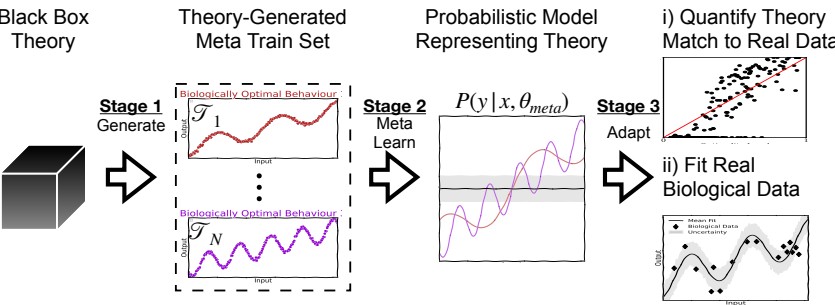

Figure 1: **Overview of the Proposed Framework**.

## 1 INTRODUCTION

A striking feature of living matter is *teleonomy*, or apparent goal-directedness. This idea permeates quantitative life sciences as a predictive principle: by assuming organisms evolve under selection for functional phenotypes, we can model their structure and behaviours as optimized solutions to ecologically relevant tasks (Bialek, 2012). This longstanding tradition, often called *normative*, or more recently *task-driven* modeling, has deep roots in computational neuroscience and sensory coding theory (Barlow et al., 1961; Atick, 1992), having provided a *top-down* explanation for the structure and organization of receptive fields in early visual areas such as the retina and primary visual cortex (Olshausen & Field, 1996; Karklin & Simoncelli, 2011). It has also proven directly

predictive of activity in higher visual, somatosensory, and auditory areas - often outperforming purely data-driven models (Yamins et al., 2014; Kell et al., 2018; Zhuang et al., 2021; Vargas et al., 2024).

Despite mounting evidence demonstrating the broad predictive and explanatory power of top-down models, this enterprise suffers from two important challenges. **First**, given multiple competing normative theories of the same system, there is no general, principled procedure in neuroscience to quantitatively arbitrate between them. Solutions like Bayesian Model Selection (Kass & Raftery, 1995) do exist, but are challenging to apply due to their requirement of first hand-specifying a probabilistic model representing the theory knowledge, and the typical intractability of the resulting marginal likelihoods, which are difficult to even estimate. **Second**, despite calls in scientific machine learning for domain-informed inductive biases, it remains difficult to systematically combine the theoretical knowledge of a good top-down model with the noisy, complex reality of empirical data to aid data-driven predictive inference. Current approaches to solving these challenges require laborious expert hand-design, and are either somewhat heuristic and system-specific (for example, constraining the receptive fields of models of early visual systems (Qiu et al., 2023b; Goldin et al., 2023)) or limited in applicability to idealized simple models (Młynarski et al., 2021).

Here, we identify Bayesian meta-learning as a principled methodological solution to both problems. We present a framework designed to automatically construct theory-informed probabilistic models applicable to any system for which top-down theories that can generate input/output predictions are available. We demonstrate our framework using a bespoke variant of adaptive deep kernel meta-learning (Patacchiola et al., 2020; Chen et al., 2023): meta-learning a Gaussian process kernel from synthetic data generated by the well-known *efficient coding* normative theory of the retina. Concretely, we first meta-learn a deep feature extractor shared across synthetic tasks derived from efficient coding theory. By virtue of meta-learning, this feature extractor forms an abstract metric embedding where the geometric distances between transformed inputs are meaningful for the *entire class* of functions consistent with our instantiation of the theory. The frozen embedding is subsequently used and updated by downstream task-adaptive components to form a *Theory-Informed Kernel* for biological data. Our informed kernel defines a Gaussian process prior over the space of functions, or an *inductive bias*, representing the theorized biological purpose of the system.

We show that this theory-informed inductive bias outperforms conventional data-driven baselines for learning the neural encoding function of *ex vivo* retinal ganglion cells, providing evidence that our framework fruitfully transfers knowledge from the theoretical model to real biological data. Further, the probabilistic theory-informed models produced by our framework natively allow the exact computation of the Marginal Likelihoods needed for Bayesian model comparison, thereby enabling rigorous information-theoretic quantification of the match between theoretical model and real data. We show that this is sensitive enough to accurately infer the *degree* of match between theory and data, which is more challenging than a simple binary choice between competing models. All in all, our work provides a much needed bridge between top-down theory-driven understanding, and bottom-up data-driven prediction (Breiman, 2001).

**Our main contribution** is a framework enabling new scientific capabilities for neuroscience that is built upon and contributes to recent advances in Bayesian machine learning. **For neuroscience**, we introduce a path to automatically construct tractable probabilistic models from "black-box" theories (Figs. 1,2), enabling the rigorous, information-theoretic validation of competing scientific hypotheses (Fig. 5) in a way that is insensitive to the theory's parameterization. Our theory-informed model improves predictive accuracy and uncertainty quantification on real data (Fig. 3), while also providing interpretability and allowing the embedded theory knowledge to be automatically relaxed or enriched as the data demands (Fig. 4). **For Bayesian machine learning**, we provide a compelling new application of deep kernels for the open problem of domain-informed kernel design. We demonstrate a successful real-world implementation: introducing a bespoke architecture and other practical design choices useful for future work (S.F.4) that avoid the common "feature collapse" pathology (Ober et al., 2021) to deliver improved uncertainty quantification (Fig. 3b,c). Finally, we validate key Bayesian deep learning principles (Wilson & Izmailov, 2020) relating generalization to inductive biases (Fig. 3a) on a challenging scientific problem.

## 2 BACKGROUND

**Top-Down Theories in Neuroscience.** *Normative theories* (also known as optimality theories)

propose that the properties of biological systems emerge as a result of evolutionary pressures optimizing a utility functional. In these frameworks, the parameters governing neural coding are regarded as optimal solutions to the underlying ecological tasks. Formally, one may express this as $\theta^* = \arg\max_\theta U\big[P(r|x,\theta), P_x(x)\big]$ where $r$ is the neural response, $x$ denotes the stimulus drawn from a naturalistic distribution $P_x(x)$, $\theta$ are the model parameters, and $U$ a carefully designed utility functional that quantifies how well the response meets the system's theorized ecological or behavioral goals. Of particular relevance is the development of efficient coding theories in early sensory systems (Barlow et al., 1961; Atick, 1992; Olshausen & Field, 1996; Karklin & Simoncelli, 2011; Karklin & Lewicki, 2009; Attneave, 1954; Simoncelli & Olshausen, 2001; Lewicki, 2002; Hyvärinen et al., 2009). This line of research was among the first to derive the structure of retinal and primary visual cortical encoding by assuming that it optimizes information transmission using limited neural activity. Contemporary *task-driven* extensions operationalize these normative principles by optimizing flexible models, often deep neural networks, directly on ecologically relevant task performance, proving highly predictive of neural activity across various domains (Yamins et al., 2014; Kell et al., 2018; Vargas et al., 2024).

**Gaussian Processes (GPs).** GPs define distributions over functions such that any finite set of points drawn from the process has a joint Gaussian distribution. In machine learning, GPs serve as Bayesian priors for functions $f$ in regression and classification models (Williams & Rasmussen, 2006; MacKay, 2003). Concretely, we place a GP prior $f_x \sim \mathcal{N}(0, K_{\theta_{\mathrm{gp}}}(\cdot,\cdot))$ with a positive-definite kernel $K_{\theta_{\mathrm{gp}}}$, and link $f_x$ to observed data $y$ through a Gaussian likelihood for regression: $y(x) = \mathcal{N}(f_x, \sigma_\eta I)$. To compute the final predictive equations of all GP's in this work, GP priors are conditioned on training data $\mathcal{D}_{\mathrm{train}} = \{X, y\}$ via Bayesian inference: $P(y_*|X_*, \mathcal{D}_{\mathrm{train}}) = \mathcal{N}\big(K(X_*, X)[K(X, X) + \sigma_\eta^2 I]^{-1}y, \quad K(X_*, X_*) - K(X_*, X)[K(X, X) + \sigma_\eta^2 I]^{-1}K(X, X_*)\big)$, where "$*$" denotes "test data", and $K(X_1, X_2)$ is a matrix shaped $|X_1| \times |X_2|$. The kernel entirely specifies the inductive bias, since it determines the mean and covariance functions of the underlying Gaussian prior. **Deep Kernels** improve GP expressivity by parameterizing the kernel with a neural network (Wilson et al., 2016). Hyperparameters $\theta_{\mathrm{gp}}$ of the kernel and likelihood can be selected by maximizing the marginal likelihood of the data:

$$P(y|x,\theta) \equiv \int P(y|f)p(f|x,\theta)df. \tag{1}$$

**Meta-Learning.** The goal of meta-learning is to use labeled input/output data from tasks in a *meta-train set* $\mathcal{T} = \{\{x_j, y_j\}_{j=1}^{n_i}\}_{i=1}^{N_\mathcal{T}}$ (where $N_\mathcal{T}$ denotes the number of tasks, and $n_i$ the number of data points per task) to learn inductive biases that are useful for fitting on other tasks in the *meta-test set* (Caruana, 1997; Zhang & Yang, 2018). Individual tasks are split into *support* and *query* data (i.e. $n_i = n_i^s + n_i^q$), where the support set is analogous to the training data for that task, and query to the validation data. Meta-learned models may contain task-adaptive components, which (unlike fixed meta-learned components) can be re-fit to every individual task, often on different optimization objectives.

## 3 Theory-Informed Meta-Learning Framework

The framework's application has three steps (Fig. 1): generating synthetic data from (normative) theory, meta-learning a theory-informed kernel from the synthetic data, and adapting/applying the resulting GP prior to real data.

Our Theory-Informed Kernel comprises disjoint meta-learned and task-adaptive modules (Fig. 2):

- A meta-learned high-capacity feature extractor $\phi$, shared across tasks, that learns an abstract metric embedding where the geometric distances between inputs are meaningful for the entire class of theory-consistent functions

- A task-adaptive linear head (per task) , $\{h_i\}_{i=1}^{N_\mathcal{T}}$, that adapts the initial metric embedding. Here, the $\{h_i\}$ serve two important purposes. First, by enabling task-specific adaptations of the metric embedding they help mitigate negative transfer (Standley et al., 2020) - alleviating the scenario where different tasks have slightly incongruent structure, thereby requiring customized embeddings. Second, the additional flexibility helps adapt the structure learned

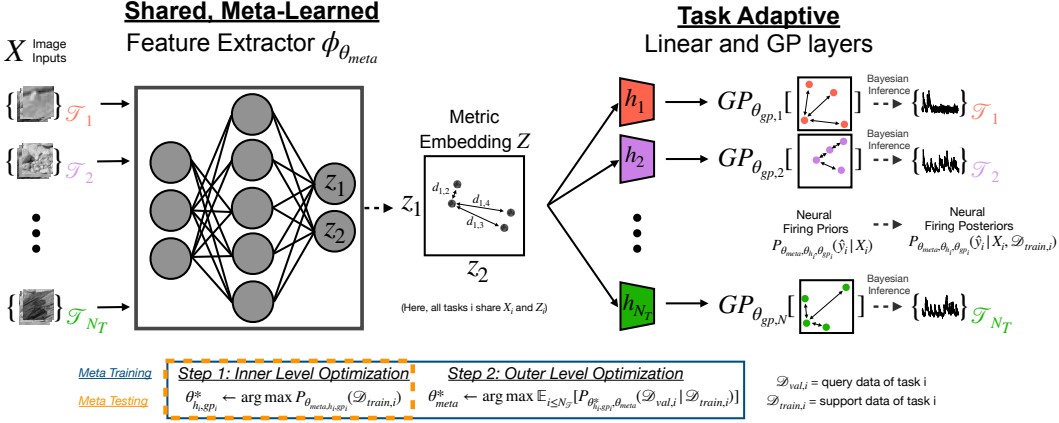

Figure 2: **Task-Adaptive Deep Kernel Meta-Learning Approach**. Inputs (e.g., images) pass through a shared feature extractor $\phi$ whose weights $\theta_{\text{meta}}$ are meta-learned across tasks. The resulting features are passed to a linear head $h_i$ and ultimately the Gaussian process (parameters $\theta_{\text{gp},i}$) layer, both of which are adapted separately to every task $i$ for which we generate predictions (e.g., neural activity). Together, these components constitute the Theory-Informed Kernel.

entirely on synthetic data to the idiosyncrasies of real biological measurements- thereby bridging the gap between simulated and real data.

- A task-adaptive Gaussian process layer (per task) that makes predictions by conditioning the theory-informed prior on the transformed inputs. Here we use the *Radial Basis Functions* kernel $K_{\text{RBF}}(z, z'; \theta_{\text{gp}}) = \sigma_f \exp\left(-\frac{|z-z'|^2}{2\ell^2}\right)$, with hyperparameters $\theta_{\text{gp}} = \{(\sigma_f, \ell, \sigma_\eta)_i\}_{i=1}^{N_{\mathcal{T}}}$ (denoting the output scale, length scale, and likelihood noise level respectively).

The meta-training procedure (Algorithm 1) employs bi-level optimization akin to (Chen et al., 2023): first an inner loop fits the task-specific linear head ($h_i$) and GP hyperparameters ($\theta_{\text{gp},i}$) using support data for each task by optimizing Eq. 1, then an outer loop updates the shared feature extractor ($\phi$) based on the adapted model's log-likelihood on query data. Once trained, $\phi$ is frozen, while task-adaptive modules freshly adapt the prior to any meta-test task (S.F.3, Eq 1). **The final Theory-Informed Kernel** (TIK) specifying the prior for each task $i$ is thus $K_{\text{TIK},i}(x, x') = \sigma_{f,i} \exp\left(-\frac{|h_i(\phi(x)) - h_i(\phi(x'))|^2}{2\ell_i^2}\right)$.

## 4 EXAMPLE: THEORY-INFORMED KERNEL FOR EFFICIENT CODING IN THE RETINA

We demonstrate our framework on mouse retinal ganglion cell (RGC) responses to natural images.

### 4.1 BIOLOGICAL DATA AND PREPROCESSING

We gratefully use previously published *ex vivo* calcium imaging data from 86 RGCs of a single mouse retina responding to 50-frame natural movie sequences presented at 30Hz in UV channels (Qiu et al., 2023a). Inputs are derived from downsampled $36 \times 32$ pixel movie frames. A single measured fluorescence response (output $\in \mathbb{R}$) is recorded per 50 preceding movie frames for each neuron, as a proxy for that neuron's activity (S.D). Each neuron's responses to all stimuli constitute a separate task, and we use the terms *neurons* and *tasks* interchangeably throughout this section. The dataset came pre-split (16200/750 train/test+val) and prepreprocessed. To adapt the data for image-based models, we temporally flattened the movies using a canonical temporal filter derived from Linear-Nonlinear (LN) model fits to the neural data (S.D.2). We then eliminated near-duplicate images via hierarchical clustering (S.D.3), yielding (1452/(400,350) train/(test,val)) unique images $\in \mathbb{R}^{36\times32}$. We therefore have **86 biological meta-test tasks**, each corresponding to the responses ($\in \mathbb{R}$) of one biological neuron to 1452 natural images $\in \mathbb{R}^{36\times32}$. From a machine-learning perspective, this corresponds to

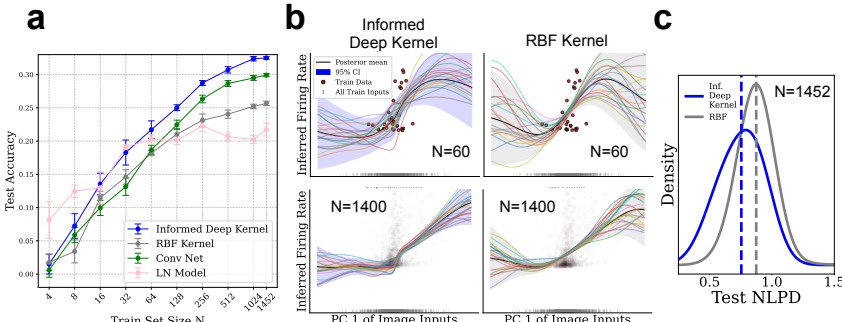

Figure 3: **Theory-Informed Kernel improves fit on retinal data.** **(a)** Test accuracy of model predictions at varying train set sizes $N$ (mean Pearson correlation over 86 neurons, standard error over 5 seeds). **(b)** Posterior mean (solid line), epistemic uncertainty (shaded), posterior samples (colored lines) of tested GP models over the first principal component (PC1) of train images. Models conditioned on $N = 60$ (top) and $N = 1400$ (bottom) points (red dots). Projected full training images shown as gray ticks (above x-axes). **(c)** Kernel Density Estimate (KDE) of the NLPD scores ; lower is better) of the RBF and the Theory-Informed kernels (all 86 neurons, 5 seeds).

86 scalar-output regression tasks, each with 1452 input–output pairs $\{x_n, y_{i,n}\}$, where $x_n \in \mathbb{R}^{36 \times 32}$ is the $n$́h preprocessed image and $y_{i,n} \in \mathbb{R}$ is the measured response to image $n$ of neuron $i$.

## 4.2 Constructing the Theory-Informed Meta-Train Set

To synthesize the meta-train set, we adapted the efficient coding (EC) normative model of Ocko et al. (Ocko et al., 2018). This model is a convolutional autoencoder trained on natural images with the EC objective of maximizing reconstruction accuracy while minimizing activity in the autoencoder bottleneck layer given constraints. A single synthetic meta-train task, $\mathcal{T}_i$, corresponds to learning the linearized encoding function of a single optimized bottleneck neuron. Given a set of N natural images ($x_k$, $36 \times 32$ pixels), each task is to predict the $N$ corresponding scalar responses, defined by the dot product $y_{i,k} = f_i^T x_k$, where $f_i$ is the neuron's specific receptive field (RF). The linearization isolates the well-validated (Hyvärinen et al., 2009; Karklin & Simoncelli, 2011) center-surround RFs that emerge as the single-neuron consequences of the population-level efficient coding objective. We extracted these RFs ($f_i$) from the optimized bottleneck by fitting an LN model to its activations. We then applied standard data augmentation and preprocessing to extend the meta-train set, resulting in a final **490 synthetic meta-train tasks derived from EC theory**. Full details of this procedure, including rationale and sensitivity analysis, are provided in S.E

## 4.3 Theory-Informed Kernel Improves Learning

Following Sec. 3, we meta-learn the feature extractor ($\phi$) on our theory-generated data (Sec. 4.2). Freezing $\phi$, we fit the task-adaptive components ($h_i$ and $\theta_{\text{gp},i}$) for each neuron $i$ ($N_{\mathcal{T}}^{\text{test}}$= 86) using Eq. 1. To assess the transfer of our meta-learned inductive bias, we then compare performance metrics against standard neuroscience baselines. Our first baseline is the well-validated *Systems Identification* convolutional neural network (CNN), specifically designed for this task (Qiu et al., 2023b). Our second baseline is the standard *Radial Basis Functions* (RBF) GP, with kernel parameters set by optimizing Eq. 1. We also compare against the Linear-Nonlinear (LN) model, transfer learning, and ablated versions of our own model, designed to ensure that the transferred features reflect knowledge obtained from the theory (S.B). We quantified the test accuracy using the Pearson correlation between predicted and measured neural responses. Hyperparameters were identified via cross validation; early stopping was used for our validation data (350 samples), equally for all models (S.G.2).

Fig. 3a depicts the performance of our model alongside the baselines, as a function of the number of training data points $N$. The experimental results follow clear trends, with the RBF GP performing competitively with the CNN at low data sizes ($N \leq 64$), and the LN outperforming the Theory-Informed Kernel in extremely low data regimes, where it is difficult to learn even the linear heads ($N \leq 8$). Nonetheless, across a large range in $N$, we find that our model outperforms all baselines

(Fig. 3a, S.B). Because our approach ultimately feeds the data embeddings obtained by the meta-learned feature extractor $\phi$ and linear heads $h_i$ into its own RBF GP, the improved performance over the bare RBF GP operating directly on the image inputs indicates that the learned embedding is relevant to modelling the structure in the biological data. Because the Theory-Informed Kernel significantly outperforms ablated versions where we either randomize or remove the meta-learned $\phi$ (S.B) and the meta-train set, we conclude that $\phi$ indeed encapsulates knowledge from our instance of efficient coding theory, thereby improving performance by a higher margin than learning the task-adaptive parameters alone, or simply meta-learning on a generic task would allow.

Comparatively, our model achieves both high data efficiency for small $N$, as well as the highest accuracy using the large $N$ (full dataset). This supports the contemporary narrative in probabilistic deep learning: that well-calibrated but flexible inductive biases are key to powerful models that perform well with any amount of data (Wilson & Izmailov, 2020).

### 4.4 THEORY-INFORMED KERNEL PRESERVES EPISTEMIC UNCERTAINTY

A crucial property of probabilistic methods like ours is their ability to represent uncertainty in their predictions. In uncertainty quantification (UQ), an important distinction is made between *epistemic* and *aleatoric* uncertainties (Hüllermeier & Waegeman, 2021), with the former being unpredictability reducible by observing more data or improving model specification, and the latter being irreducible unpredictability (e.g., noise). Due to pathologies that arise from the interaction between deep learning and the marginal likelihood objective (Eq. 1), Deep Kernel GPs notoriously tend to exhibit *feature collapse*, specifically resulting in overconfident epistemic uncertainty (Van Amersfoort et al., 2021; Liu et al., 2020; Ober et al., 2021). Naturally, this is a concern for our framework.

To assess whether our model's representation supports reasonable uncertainty quantification, we compare it to the RBF GP, with both models fit on the full train set. We first present, on the best fit neuron for both models ($i = 37$), visualizations of their predictions and 95% CI (here only epistemic uncertainty), as a function of the first principal component (PC1) of the full stimulus train set (fixing the other PCs to their means). We show cases where both models are conditioned on either $N = 60$ or $N = 1400$ data (Fig. 3b). Both models exhibit the characteristic reduction of uncertainty near the observed data at $N = 60$, but unlike the RBF GP, our Theory-Informed Kernel heteroskedastically preserves some uncertainty near PC1 = 0 where the samples are dense. Conditioned on $N = 1400$ data, the epistemic uncertainty of the RBF GP collapses near the data, while the informed kernel maintains a reasonable confidence region enveloping a substantial proportion of the data (despite only showing the epistemic part). This suggests, qualitatively, our representation improves uncertainty representation, reducing overconfidence in regions with many observations.

Next, to quantitatively assess UQ in our model, we compute the standard Negative Log Predictive Density (NLPD) metric on the full images: $\text{NLPD}(y_{\text{test},i}|\text{model}_i) \equiv -\log P(y_{\text{test},i}|x_{\text{test},i}, x_{\text{train},i}, y_{\text{train},i}, \phi, \theta_{\text{gp},i}, h_i)$ For neuron $i$, $P(y_{\text{test},i}|\cdot)$ denotes the GP posterior predictive distribution obtained by conditioning $\text{GP}_i$ on $\mathcal{D}_{\text{train},i}$. Evaluating this posterior at $M$ test inputs yields an $M$-dimensional multivariate Gaussian predictive distribution $\mathcal{N}(\mu_i, \Sigma_i)$, and the NLPD is the negative log-probability of the observed test responses under this Gaussian. Lower NLPD indicates more probability mass on the true responses, reflecting both mean accuracy and uncertainty calibration. For this metric, we consider both the epistemic and alleatoric uncertainties. Fig. 3c shows the kernel density estimate of NLPD values across all neurons and data. Our Theory-Informed Kernel exceeds the UQ capabilities of the RBF baseline on biological data, avoiding common Deep Kernel GP pitfalls (Ober et al., 2021) and enabling potential downstream applications where it is vital for the model to "know what it doesn't know" (Cowley et al., 2017; Frazier, 2018).

### 4.5 THEORY-INFORMED KERNEL LEARNS INTERPRETABLE REPRESENTATIONS

How exactly does our model's learned representation support its predictive performance? As a Deep Kernel GP with an RBF layer, its predictions are driven by the pairwise Euclidean distances between transformed inputs. To gain mechanistic insight into how the task-specific linear transformations $h_i$ tailor the model's function, we analyze their impact on these critical pairwise distances. Starting with a set of probe images $\pi$, we compute a *prototype image*, $\mathcal{P}_i$, for each $h_i$, by summing the pixel-level overlap masks between pairs of images $x_m, x_n \in \pi$ weighted by how $h_i$ alters their

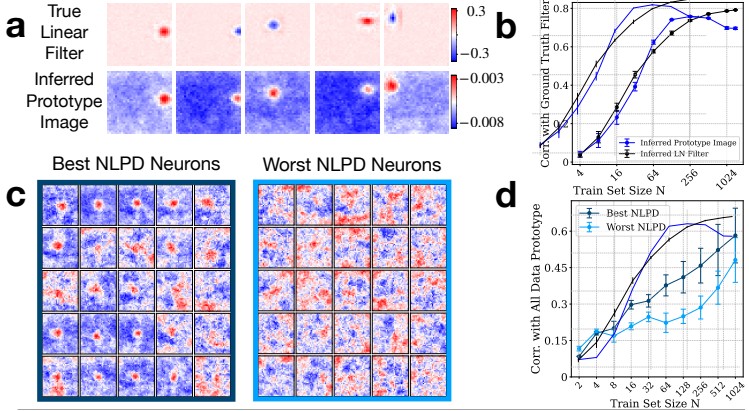

Figure 4: **Task-Adaptive Heads Learn Interpretable Representations. (a)** Prototype images ($\mathcal{P}_i$) derived from learned representations vs. ground truth linear filters of synthetic neurons. **(b)** Mean Pearson corr. between the absolute values of the $\mathcal{P}_i$ (blue) and filters learned by an LN model (black) with ground truth as in (a), varying train set size $N$. **(c)** $\mathcal{P}_i$ for the 25 best (dark blue) and worst (light blue) performing real neurons, sorted by NLPD, learned from the full dataset ($N = 1452$). **(d)** Mean Pearson corr. between $\mathcal{P}_i$ at varying $N$, with the $\mathcal{P}_i$ at full dataset ($N = 1452$), split according to NLPD as in (c). (b) and (d) averages over neurons with standard errors across 5 random seeds.

projected distances $\|h_i(\phi(x_m)) - h_i(\phi(x_n))\|_2$ (S.H.1). These $\mathcal{P}_i$ represent features of the inputs that $h_i$ learn to 'pull together' or 'push apart' for their task $i$, in their output representation distances.

We first inspect the prototype images on synthetic data, finding that prototype images $\mathcal{P}_i$ recapitulate the filters used to generate the theory-informed responses very well (Fig. 4a). We further observe that the rate at which the prototype image of the Theory-Informed Kernel recovers (in absolute value) the ground truth RF filter as a function of train set size $N$ closely matches the rate of recovery by the LN model's linear filter (Fig. 4b), indicating the importance of $\mathcal{P}_i$ for this model. At high train set size $N$, however, the correlation between the ground truth and prototype image stops increasing monotonically, in a textbook example of the *Bayesian Occam effect* (Rasmussen & Ghahramani, 2000; MacKay, 1992; 2003). As $N$ grows, the GP layer's non-parametric capacity automatically expands, so the marginal likelihood stops rewarding additional structure in $h_i$ and favors a less informed prior. This contextualizes our argument from Sec. 4.3 about *informed yet flexible* models that retain their advantage across all dataset sizes: when data are scarce, the theory-informed inductive bias supplies the needed structure; when data are plentiful, Eq. 1 automatically relaxes that structure to permit a predictive fit.

Applied to biological data, we find a clear distinction between $\mathcal{P}_i$ corresponding to tasks $i$ on which the model performed well, and those on which it performed poorly. The 25 best performing (by NLPD) $h_i$ transform the data in interpretable ways: resembling what may be the receptive fields of the biological neurons (Fig. 4c). The 25 worst $h_i$, conversely, seemingly fail to do so. This is reflected in (Fig. 4d), which depicts the Pearson correlation between the prototype images learned at various $N$, with the prototype image given the full data; the worst performing tasks learn a less interpretable and also less consistent prototype image. Unlike the synthetic case, for real neurons the structure in $\mathcal{P}_i$ keeps increasing up to $N = 1024$, indicating that we are still in a prior-aided regime where the data alone have not saturated the non-parametric capacity.

Taken together, our Theory-Informed Kernel framework is both *performant* and *interpretable*. When it fails, we gain insight into why; when it succeeds, we can watch the Occam effect unfold, demonstrating that Bayesian principles can deliver practical gains in real-world settings. Moreover, our prototype-based interpretation is readily transferable to domains beyond sensory neuroscience, wherever one wishes to visualize what a deep-kernel GP has learned to look for.

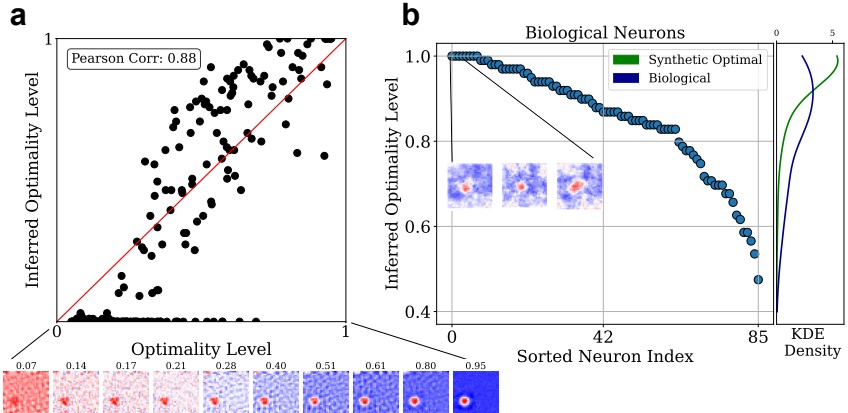

Figure 5: **Theory Informed Kernel quantifies Theory Fidelity. (a)** Degree of optimality, $\beta^*$, inferred by our framework, plotted against ground truth on synthetic neurons (black dots; Pearson correlation = 0.88). Subpanel: example synthetic receptive fields of varying optimality. **(b)** Scree plot with the inferred efficient coding level of optimality $\beta^*$ for all 86 recorded retinal ganglion cells. Inset: "Prototype images" $\mathcal{P}$ of top three RGCs (detailed in Fig. 4). Right margin: KDE of inferred optimality for all 86 RGCs (blue) and 100 synthetic optimal neurons (green).

### 4.6 THEORY-INFORMED KERNEL ENABLES VALIDATION OF BLACK BOX NORMATIVE THEORIES

A major methodological challenge in top-down (either normative or task-driven) modelling is the validation of how well a hypothesized optimality theory describes the biological system. Here, we cast the problem of quantifying the degree to which an optimality theory describes the real dataset as *Bayesian Model Comparison* (BMC). Typically, BMC involves comparing the marginal likelihood, $P(\mathcal{D}|\mathcal{M})$, which quantifies the evidence for competing hypotheses (models $\mathcal{M}$) given the data $\mathcal{D}$. A Bayesian might decide between $\mathcal{M} = 1$ (e.g., a specific theory) and $\mathcal{M} = 0$ (e.g., a null hypothesis) by selecting the highest scoring $\mathcal{M}$. In our GP framework, this suggests a comparison between the theory-informed kernel and a strong yet theory-agnostic *null model* kernel like the RBF. However, a simple binary choice between these two alternatives may be inadequate for biological systems, which often only partially adhere to the postulated theoretical principles (e.g., neurons sometimes exhibit deviations from the theoretically-predicted center-surround receptive fields). We thus propose $K_\beta$, interpolating between theory-informed (TIK) and theory-agnostic (RBF) kernels:

$$K_\beta(x, x') = \beta K_{\text{TIK}}(x, x') + (1 - \beta)K_{\text{RBF}}(x, x') \tag{2}$$

Since the marginal likelihood quantifies the evidence for the specific prior defined by $K_\beta$, the optimal $\beta^* = \arg\max_\beta P(Y|X, \beta)$ indicates the degree to which the data provides evidence for the theory-informed structure beyond what can be captured by the powerful-yet-generic null kernel. In other words, $\beta^*$ can be interpreted as the inferred measure of the theory's relevance, or alternatively, the system's *degree of postulated optimality*.

We first validate this approach on synthetic tasks. We generate these by starting with 20 optimal synthetic receptive fields (RFs) unseen during meta-training, and gradually add structured noise to push them away from optimality (S.I.4). As a proxy for the ground truth optimality of these new synthetic RFs, we use the standard procedure of computing the $R^2$ of a fit of the *Difference-of-Gaussians* (DoG) model to the RF, as DoG are theorized to constitute the efficient coding solution for the retina (S.I.4) (Atick & Redlich, 1992). We then freeze the feature extractor $\phi$ and fit the task-adaptive parameters (linear heads $h_i$ and GP parameters $\theta_{\text{GP},i}$) for $K_{\text{TIK}}$, as well as the parameters $\theta_{\text{GP},i}^{\text{null}}$ of $K_{\text{RBF}}$, separately on each task $i$. Finally, with all parameters frozen for both kernels, we find optimal $\beta_i^* = \arg\max_{\beta \in [0,1]} P(Y_i|X_i, \beta)$ via grid search over 100 points. Fig. 5a shows that $\beta^*$, inferred neuron by neuron as described, correlates strongly with the ground truth optimality level on synthetic data. Given a postulated normative theory, $\beta^*$ therefore approximates the neuron's *degree of optimality*. We validate the importance of our theory-informed features with an ablation study in S.I.

Next, we repeat the same procedure on our biological RGC dataset, by estimating $\beta_i^*$ for each neuron $i$. The inferred optimality levels of all 86 neurons are presented in Fig. 5b, as an example of a scientifically relevant analysis enabled by our Bayesian framework. Most neurons score highly on the proposed optimality metric $\beta^*$, consistent with the general consensus about the applicability of the efficient coding theory (Simoncelli & Olshausen, 2001; Pitkow & Meister, 2012). For comparison, Fig. 5b also depicts, in the right margin, the $\beta^*$ of 100 randomly selected synthetic optimal neurons (unseen in meta-training).

The ability of our framework to quantify how well a top-down theory explains neural data is a practical step forward for neuroscience. Though BMC has long been advocated as the principled solution to this problem (Jaynes, 2003), its routine use has been hindered by two obstacles: (i) manually translating a normative theory into a predictive probabilistic model; and (ii) computing the resulting marginal likelihoods in high-dimensional neural datasets. Consequently, most studies still rely on heuristic comparisons of summary statistics, such as the receptive field structure or tuning curve parameters (Ringach et al., 2002; Karklin & Simoncelli, 2011). By embedding normative theories into a Gaussian process framework and leveraging the exact tractability of Gaussian marginal likelihoods, our approach addresses both obstacles. This makes principled Bayesian model comparison more feasible at scale and lowers the barrier for quantitatively testing normative theories on neural data.

## 5    DISCUSSION AND CONCLUSION

**Related Work.** Concepts of meta-learning and domain-informed inductive biases pervade science, and scientific machine learning (Karniadakis et al., 2021; Lake et al., 2017; McCoy et al., 2020; Dorrell et al., 2023; Li et al., 2023). Our framework was inspired by Linderman & Gershman (2017)'s call to embed high-level computational goals of neural systems into statistical models. In line with their Bayesian tenets, where theory (via priors) is combined with data (via likelihoods), our work delivers on their promises: enhancing interpretability and identifiability, and enabling empirical validation of theories. We also build on the elegant framework of Młynarski et al. (2021), who incorporate maximum entropy priors derived from top-down theories into parametric Bayesian models, thereby improving inference and validating optimality theories using BMC. Similarly, Bittner et al. (2021) use normalizing flows to represent the distribution of mechanistic model parameters supporting desired emergent computational properties. In contrast, our approach sidesteps the need to commit to any specific parametric form, by meta-learning the priors directly in function space, yielding increased generality and scalability without compromising Bayesian principles. We are indebted to the work of Qiu et al. (2023b), who use efficient coding as an inductive bias for predictive modelling on their RGC dataset which we use here as well. Their approach hinges on enforcing shared spatial filters for two CNN models trained in parallel: one performing efficient coding (EC), the other fitting data, yielding improved accuracy and biologically plausible CNN filters. Unlike Qiu et al, who specifically design an inductive bias to exploit EC's ability to produce useful spatial filters for a CNN, our framework does not assume the biological system to be modeled or the theory knowledge to be meta-learned, thus promising applicability to a broader class of systems. Our GP architecture is a variant of Chen et al. (2023) meta-learner, to which we have introduced the $h_i$ (crucial to bridge the sim-to-real gap), and adjusted the training to avoid feature collapse (S.F.4). Architecture aside, our work fundamentally differs: focusing on theory to data transfer rather than data-driven meta-learning. Finally, our framework is related to contemporaneous work by McCoy & Griffiths (2025), who also use meta learning on synthetic data to distill inductive biases from theoretical models. In their domain of linguistics, the purpose of their meta learning component is to take Bayesian theoretical models' inductive biases and embed them within a non-Bayesian neural network model for both few-shot learning and flexible learning on complex naturalistic tasks. Like in our framework, they show inductive bias transfer conferring improved learning. Unlike ours, their primary motivation is to embed initially Bayesian theories into flexible non-Bayesian neural networks, while ours conversely is to take black box models and distill their predictive structure *into* a Bayesian prior. We elaborate in (S.A), particularly contrasting our work with that of Qiu and Młynarski.

**Weaknesses and Future Directions.** **(i)** A way to inspect *what* meta-learned knowledge is actually stored in the shared feature extractor $\phi$ would be very useful, especially for domains other than efficient coding where predictive aspects of normative theories are less well understood. Although this is a generic limitation of meta-learning, the inherent interpretability of deep kernel representations, demonstrated in Sec. 4.5, may provide a stepping stone for future work. **(ii)** In seeking a principled

separation between meta-learned knowledge and task-specific adaptations, we deliberately used a frozen, meta-learned feature extractor $\phi$ in our kernel. An exciting open question is whether the knowledge in $\phi$ could be enriched by carefully allowing its adaptation also on biological data. **(iii)** Predictions relating to Bayesian Model Comparison suffer from all of the commonly known challenges of using BMC with powerful models (Kass & Raftery, 1995; Gelman et al., 1995; Bernardo & Smith, 2009): namely, sensitivity to data properties, to the null model, and to the adaptation of the prior. Finally, **(iv)**, though our framework is by design general, our empirical validation is limited to a single challenging system. We justify our claim to generality in (S.C).

While there are many prospective improvements that are interesting research directions, we view them as promising future extensions to our framework and not as refutations of its contributions.

**Conclusion.** We have presented a Bayesian framework to meta-learn flexible probabilistic models from top-down theories. Using deep kernel GPs, we've successfully transferred knowledge from an efficient coding theory of early visual processing to model *ex vivo* neural recordings, demonstrating interpretability and improving accuracy, uncertainty quantification, and data efficiency compared to baselines and ablations. Through Bayesian model comparison, our framework provides a new scalable path to quantitatively validate normative theories against empirical data—an advancement with far-reaching potential for neuroscience.

REPRODUCIBILITY STATEMENT

ETHICS STATEMENT

This research exclusively utilized a pre-existing, publicly available dataset of ex vivo mouse retinal ganglion cell recordings (Qiu et al., 2023b;a). No new animal experiments were conducted for this study. The original data collection was performed by the cited authors under their institution's approved protocols for animal welfare. Our work is therefore confined to the computational analysis of this existing data. Beyond the use of this dataset, our research does not present foreseeable ethical concerns.

## 5.1 ACKNOWLEDGEMENTS

This research was funded in whole or in part by the Austrian Science Fund (FWF) 10.55776/COE16. We thank Wiktor Młynarski and Henry Moss for helpful discussions. We thank Max Cairney-Leeming and James Odgers for feedback on the manuscript.

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

SUPPLEMENTARY MATERIAL: META LEARNING THEORY INFORMED INDUCTIVE BIASES USING DEEP KERNEL GAUSSIAN PROCESSES

## S.A EXTENDED RELATED WORK

Here we extend the discussion on some closely related work, showing how our framework differs.

Młynarski et al. (2021) present work related to ours- a framework to construct normative-theory informed parameter priors for parametric models. They specify their normative prior numerically using a specific utility function $U(\theta)$ for a hand-crafted parameterization in $\theta$. Their prior, $\frac{\exp[\beta U(\theta)]}{Z(\beta)}$ crucially requires computing a normalization constant $Z$ by integration, which they estimate by monte carlo but, as they note, is intractable for even 1000 dimensional cases (e.g. our RGC case). Their model comparison application also hinges on this constant (equiv. to our marginal likelihood (ML)), and therefore must be approximated. These intractabilities limit the applicability of Młynarski et al's framework to small, simple models with well understood, well parameterized normative theories. In contrast, our framework works with any theory that can generate raw predictions (e.g. task-driven networks, autoencoders), as it learns the prior in function space, not parameter space. We are not constrained by dimensionality, and compute our ML analytically. Our framework thus is more automated, general, and scalable while being similarly principled and rigorous.

While we build on the work of Qiu et al. (2023b) and have benefited greatly from their contributions, our work fundamentally differs from theirs in important ways: **First**, our model is probabilistic, providing well-calibrated uncertainty estimates (Fig. 3c). Furthermore, our normative inductive bias is adaptive to new tasks, and as we show in our analysis of the Occam's Razor effect (Fig. 4b), can even relax its influence as more data becomes available. **Additionally**, the model of Qiu et al is well designed to support one theory. Their method hard-codes constraints informed by the theory into the first-layer convolutional filters, limiting it to Retinal ganglion cell applications. Our framework's function-space prior is much more general, allowing it to model abstract knowledge that cannot be encoded in a first-layer spatial filter (As discussed in S.C). **Finally**, through its use of exact marginal likelihoods, our framework provides a principled, information-theoretic tool to quantitatively arbitrate between competing and structurally different theories. The closest thing Qiu et al's model can do is enable quantitative comparison of learned receptive fields- which, while valuable, is only a useful heuristic limited in applicability to this one system.

**Other related works** The work of Shrinivasan et al. (2023) shares our goal of rigorously testing normative theories with real data. However, our frameworks differ in their core approach and capabilities. Shrinivasan et al focus on manually instantiating variants of a single normative theory (the Neural Sampling Code (NSC)) as probabilistic models. They instantiate the whole generative model hypothesized by NSC $P(x, z)$ and test various competing parameterizations for the constituent $P(z|x)$ and $P(x)$, maximizing the joint likelihoods. Our work is similar in that we also principled instantiate model comparison on the basis of likelihoods of probabilistic models. In contrast to their work, we do not hand-parameterize and then fit a normative-theory inspired probabilistic model, but rather we meta-learn the probabilistic model, represented by the GP kernel function. Thus, the goal of our framework is to enable analysis like that of Shrinivasan et al but for a broader class of theories that may be difficult to hand-parameterize as a probabilistic model directly. The scientific value of Shrinivasan's work substantiates the need for a framework streamlining the process, like ours. Methodologically, Moss et al. (2024) also fit deep kernel GPs on simulated data, but focus on learning a solution operator to speed up latent force models, rather than a theory-informed kernel for data modeling. Our work is also related to that of Kobalczyk & van der Schaar (2025), who also use Bayesian meta-learning to incorporate human-interpretable domain knowledge into probabilistic models. Unlike us, the source of the knowledge in their informed neural process framework comes from an LLM-generated representation of the knowledge, (e.g. "This plot is increasing"), and they meta-learn the relationship between the knowledge embedding and the function space prior itself. Therefore their framework would not be applicable to our setting, as this form of knowledge integration would not help with validating existing theoretical models of neural systems. Finally, while our work is scoped around normative theories in neuroscience, it is related to the technique of "sim-to-real" in robotics (e.g. Peng et al. (2018)), which, similarly to us, endeavors to deploy knowledge learned from simulations to the real world. It would be interesting to see if our task-adaptive meta learning approach would help bridge the simulation to reality gap in this domain.

## S.B  EXTENDED FIGURE 3A AND ABLATION STUDY RESULTS

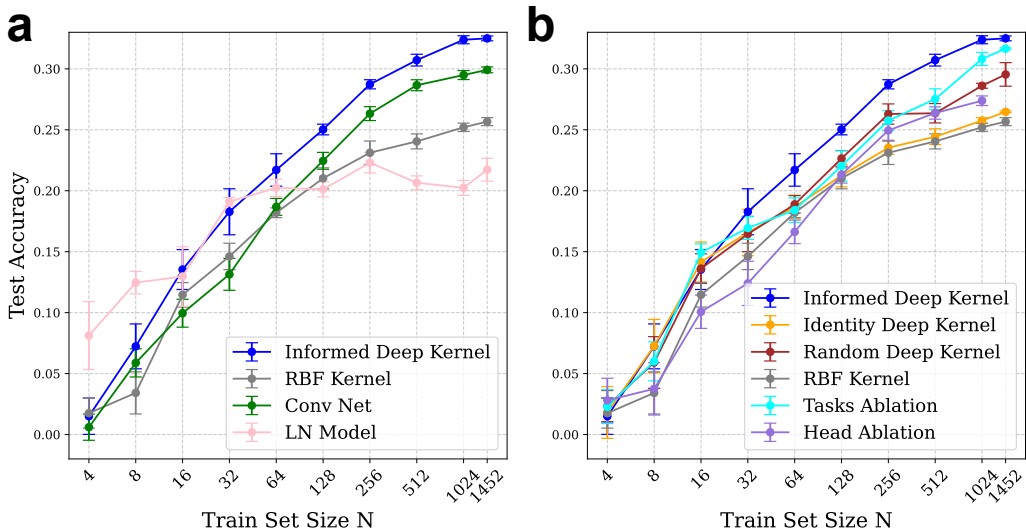

Figure S1: **Extended Figure 3a and Ablation study** (**a**) Reproduction of Figure 3a. (**b**) Accuracies of ablation models on biological data.

| $N$ | Identity | RBF | Random | TaskAblation | HeadsAblation |
|---|---|---|---|---|---|
| 4 | $2.84 \times 10^{-1}$ | $6.56 \times 10^{-1}$ | $4.34 \times 10^{-1}$ | $8.32 \times 10^{-1}$ | $8.85 \times 10^{-1}$ |
| 8 | $4.39 \times 10^{-1}$ | $1.89 \times 10^{-3}$*** | $2.67 \times 10^{-2}$** | $1.50 \times 10^{-}$ | $4.50 \times 10^{-5}$*** |
| 16 | $8.02 \times 10^{-1}$ | $3.51 \times 10^{-2}$* | $3.37 \times 10^{-1}$ | $9.40 \times 10^{-1}$ | $5.97 \times 10^{-5}$*** |
| 32 | $1.85 \times 10^{-2}$* | $1.39 \times 10^{-3}$** | $2.12 \times 10^{-2}$* | $2.23 \times 10^{-2}$* | $2.77 \times 10^{-10}$*** |
| 64 | $8.29 \times 10^{-4}$*** | $6.14 \times 10^{-3}$** | $6.77 \times 10^{-3}$** | $1.37 \times 10^{-3}$** | $1.05 \times 10^{-7}$*** |
| 128 | $1.38 \times 10^{-4}$*** | $2.35 \times 10^{-3}$** | $5.04 \times 10^{-3}$** | $2.17 \times 10^{-3}$** | $3.19 \times 10^{-5}$*** |
| 256 | $6.26 \times 10^{-6}$*** | $6.84 \times 10^{-4}$*** | $5.99 \times 10^{-3}$** | $3.87 \times 10^{-3}$** | $2.06 \times 10^{-6}$*** |
| 512 | $4.43 \times 10^{-7}$*** | $8.75 \times 10^{-5}$*** | $4.74 \times 10^{-5}$*** | $2.48 \times 10^{-3}$** | $2.98 \times 10^{-7}$*** |
| 1024 | $6.67 \times 10^{-9}$*** | $1.87 \times 10^{-5}$*** | $3.37 \times 10^{-4}$*** | $1.10 \times 10^{-1}$ | $1.15 \times 10^{-7}$*** |
| 1452 | $7.16 \times 10^{-7}$*** | $9.37 \times 10^{-5}$*** | $5.54 \times 10^{-4}$*** | $2.47 \times 10^{-1}$ | - |

Table S1: $p$-values for the one-sided paired Wilcoxon test (Hypothesis: Informed > Control) on the 86 task-level means at each training-set size. $^{*} p < 0.05$, $^{**} p < 0.01$, $^{***} p < 0.001$.

**Discussion**  Figure S1 extends Figure 3a from the main text, now including the results from our ablation studies ('Identity', 'Random', 'Head Ablation', 'Tasks Ablation').

The LN model, a common baseline for modeling retinal responses, demonstrates strong performance in the extremely low-data regime. Notably, while the LN model is broadly superior in extremely small data regimes ($N < 16$), our theory-informed model catches up in performance ($16 \leq N \leq 64$), and subsequently exceeds that of the LN model - suggesting that the transferred theoretical knowledge provides an effective bias that helps in data-limited regimes but does not hinder as data becomes more abundant.

**Ablation Studies**  To isolate the contribution of the theory-derived meta-learned feature extractor ($\phi$), we performed four ablation studies (Figure S1b):

1. An 'Identity' deep kernel ablation: This removes the feature extractor entirely, applying the task-adaptive linear heads directly on input pixels.

2. A 'Random' deep kernel ablation: This randomizes the weights of $\phi$ (sampling from the standard initialization distribution), but keeps the architecture (Table S4) fixed.

   This ablation represents a powerful comparison target, as large randomly initialized networks, particularly preceding learnable layers like our linear head and RBF GP, have been shown to represent features that capture relevant data structure, preserving input geometry, and in some cases nearly matching the performance of fully trained features ( (Rahimi & Recht, 2007; Rudi & Rosasco, 2017; Vershynin, 2018; Saxe et al., 2011; Ulyanov et al., 2018)).

3. A "Task Ablation" condition on the meta-learning process, where we retrain the model, replacing the theory-derived set of meta-training tasks with a generic tasks derived from the statistics of natural images alone. We replaced our meta-training set tasks with the tasks of trying to predict the values of the top principal components of natural images- with each task constituting one principal component. We kept all other conditions fixed.

4. A "Heads ablation" condition, where we remove the task adaptive linear heads

**Statistical tests** We quantitatively compared the performance of our informed model against these two ablations (and the RBF GP) using a one-sided paired Wilcoxon signed-rank test (Wilcoxon, 1992) (hypothesis: Informed > Control). For each training-set size $N$, we first averaged the test correlation over the five random seeds within each task, yielding 86 paired observations for the test (one per task).

The $p$-values are presented in Table S1, and statistical significance is annotated visually using asterisks in Figure S1b. Statistical significance ($p < 0.05$) emerges for all baseline comparisons at $N \geq 32$, with high significance levels ($p < 10^{-3}$) consistently observed for training-set sizes of $N \geq 64$. For reference, the median correlation improvement over random features at $N = 512$ was $0.049$ (95% BCa bootstrap confidence interval: $[0.018, 0.083]$), demonstrating that the detected differences are both statistically robust and practically meaningful.

The analysis confirms that our theory-informed model significantly outperforms the standard RBF GP baseline and the other ablation conditions across most training data sizes. The significant improvement over the 'Random' baseline is noteworthy; given that randomly initialized deep networks themselves provide strong feature representations (as evidenced by the strong performance of this baseline). The specific features obtained through meta-learning therefore confer a substantial structural advantage. Furthermore, the superior performance of both informed and random features compared to the 'Identity' baseline confirms the significant contribution of the deep feature extractor component itself, beyond the task-adaptive linear heads. Finally the superior performance of the theory-informed meta-learned features over theory-agnostic meta learned features (designed to encourage good generic features of natural images) shows that the accuracy benefit is specific to the theory-informed tasks.

**Take Away:** Our ablation study supports the claim made in the main text: the meta-learned features effectively transfer knowledge from the theory, providing a beneficial inductive bias that leads to improved performance.

### S.B.1 COMPARISON TO TRANSFER LEARNING

We here highlight why we specifically need to use meta learning for our framework, as opposed to simpler mechanisms of knowledge transfer such as transfer learning. In standard Transfer Learning (TL), prior knowledge is encoded as a static feature set. This feature set is then re-used or fine-tuned by a simple downstream predictor on another task: directly constituting part of the prediction.

Our framework operates on a different principle entirely. While there IS a frozen feature extractor involved, what we transfer is a meta-learned theory-informed Bayesian prior over the space of functions. The role of the feature extractor is **not** to provide part of the solution for a task (e.g. by learning a center surround receptive field), but to specify an intricate function space prior out of an exceptionally vast set of functions representable by an abstract metric embedding.

Intuitively, transfer learning is like memorizing a part of the "answer" for one task, and transferring it onto another task. In contrast, our meta learning framework constructs, then transfers a "scheme" for how to find the answer for any task in a given domain.

This difference is consequential, conferring the following **advantages of our Meta Learning framework over Transfer learning**:

- **Greater Generality**: As a prior over functions, our method is agnostic to exactly -what kind- of knowledge it needs to encode. While for fitting an RGC response to images, one could just as well invent clever regularizations on CNN filters to achieve similar or superior accuracy gains, this approach is limited to forms of knowledge that can be expressed as specific neural network weight values/statistics. In contrast, our prior can capture more abstract forms of knowledge, for example abstract rules for a rule-based decision making task. We discuss this more in Sec. S.C

- **Task-Adaptive Prior** Unlike transfer learning, where the knowledge transfer is fixed, our theory informed prior can adaptively relax itself or absorb more informative structure via the linear heads. This enables generalization from idealized efficient coding predictions to the real, noisy biological data.

- **Principled Theory Validation and Uncertainty Quantification** In our informed GP, we can compute the exact marginal likelihood of biological data under the theory-informed prior— a direct measure of the evidence for a given theory in units of information. This enables a direct quantitative comparison between competing scientific theories. Our GP is also probabilistic: presented with an image that its training data is not representative of, it outputs a high degree of uncertainty (Fig 3b,c), and not just a confidently wrong answer.

**Transfer Learning Baselines**  We ran an extensive grid search to fine-tune linear and MLP heads on top of frozen features from pre-trained ResNet-18 models (using both ImageNet and SimCLR trained weights). As shown in our figure , we found that these pretrained features were not competitive, performing worse than even the plain MLP on the raw data (Figure S2)

We hypothesize this is because the features from these deep networks are too abstract for RGC responses, a finding consistent with prior work ( (Cadena et al., 2019)) which found earlier-layer features were necessary even for modeling V1.

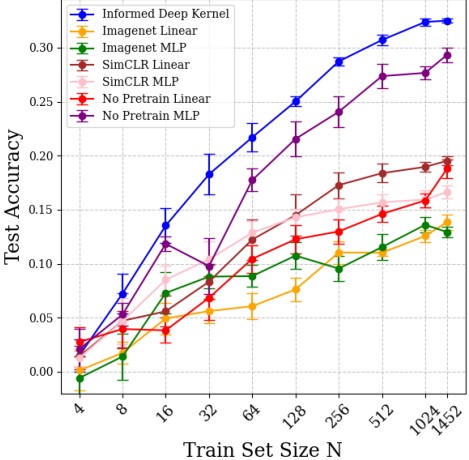

Figure S2: **Transfer Learning baselines**

S.B.2   ANALYSIS OF RELATIVE SOURCES OF INACCURACY OF DIFFERENT MODELS

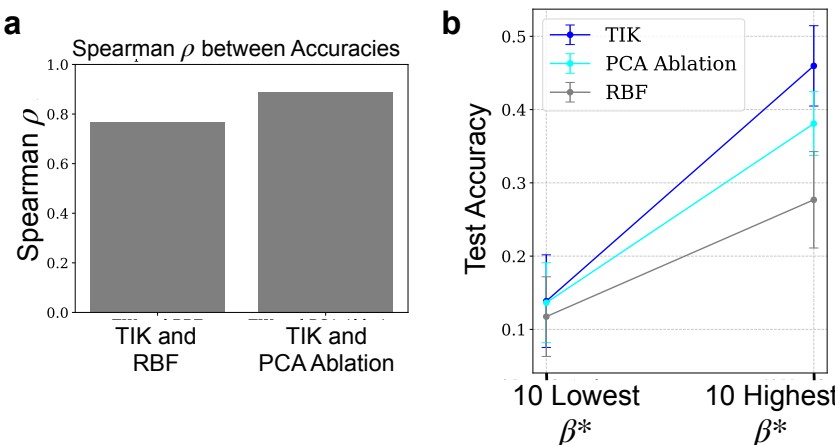

Figure S3: **Analysis of sources of error** a) Depicts Spearman rank correlation of accuracies of different methods across the 86 biological neurons, averaged over seeds. b) Depicts accuracies of TIK, PCA ablation, and RBF GP's on the 10 biological neurons with the lowest and highest $\beta^*$ scores by our BMC procedure (Fig. 5b)

Here we analyze what differentiates biological neurons on which TIK performs well from those where it performs poorly. We first compare per-neuron prediction accuracies (correlation on the test set, averaged over seeds) between models. As shown in Fig. S3a, the accuracies of TIK, the PCA ablation, and the RBF GP are strongly rank-correlated across neurons (e.g. Spearman $\rho \approx 0.77$ for TIK vs. RBF). This indicates that all three models tend to succeed and fail on the same neurons, suggesting that a substantial component of the performance variation across neurons is shared and not specific to any one kernel. We next stratify neurons by their inferred $\beta^*$ from the TIK–RBF mixture (Fig. 5b). Fig. S3b shows the mean test accuracy of each model on the 10 neurons with the lowest and highest $\beta^*$. On the 10 lowest-$\beta^*$ neurons, all three models perform similarly poorly (mean Pearson $r \approx 0.13$ for TIK, PCA, and RBF), indicating that for these cells neither the EC-informed features nor the alternative deep or RBF kernels provide a strong inductive bias. In other words, $\beta^* \approx 0.6$ corresponds to neurons that are difficult to predict for *all* models, rather than cases where the RBF null alone is a good fit. In contrast, on the 10 highest-$\beta^*$ neurons, accuracies increase for all models but in a graded fashion: relative to the low-$\beta^*$ group, the RBF GP improves by $\sim 0.16$ Pearson $r$, the PCA ablation by $\sim 0.24$, and TIK by $\sim 0.32$. Thus, neurons that our BMC procedure classifies as better explained by the theory are precisely those for which theory-informed features yield the largest additional predictive gain. The fact that the PCA ablation sits between RBF and TIK is consistent with the picture that generic natural-image structure (captured by PCA) explains part of the improvement, but that the full gain for high-$\beta^*$ neurons is only realized when using features meta-trained on the specific efficient-coding normative model.

**Implications.   Shared difficulty across models.** The high rank-correlations in Fig. S3a show that neurons which are hard for TIK are also hard for the PCA deep kernel and the RBF GP. This suggests that the dominant source of performance variation across neurons is intrinsic unpredictability of the recorded responses (due to biological variability and/or measurement noise). **Role of efficient-coding structure.** Conditioned on neurons where structure is more easily discernible (high $\beta^*$), we observe a graded improvement RBF < PCA < TIK. This pattern indicates that, beyond generic natural-image statistics, the EC-derived feature space provides additional, neuron-specific structure that improves predictive performance. **Interpretation of $\beta^*$.** High $\beta^*$ can be interpreted as identifying neurons whose responses are especially well captured by this efficient-coding solution, whereas $\beta^* \approx 0.5$ flags neurons for which **(i)** none of the tested kernels (TIK or RBF) provide a strong advantage over one another for explaining the data (e.g. due to noise), **or (ii)** the structure truly lies in between the two (e.g. in our synthetic, noise free example)

## S.C    ON THE GENERALITY OF OUR FRAMEWORK

While our manuscript presents a single challenging deep-dive case study of the retina application, our framework is, by design, quite general.

### S.C.1    DEEP KERNELS DEFINE BROADLY APPLICABLE FUNCTION SPACE PRIORS

The generality of the Deep Kernel architecture stems from a core principle: **many complex biological phenomena are governed by a small number of simple, underlying latent variables.** Systems like these can be well described by our model.

To elaborate, metric-based methods confer significant flexibility when used to define function space priors in a deep kernel Gaussian process setting. Since the core of our model is a Gaussian Process with learned kernel $K_i(x, x') = \sigma_i^2 \exp[-\frac{||h_i\psi(x) - h_i(\psi(x'))||^2}{2l_i^2}]$, the prior over functions is fundamentally defined by the geometry of the embedding space learned by $\psi$ and refined by $h_i$.

The core assumption of this architecture is that the deep neural network can learn a transformation of the potentially complex input space $\mathcal{X}$ to an embedding space $\mathcal{Z}$ where the functional relationships become simple, smooth, and stationary. The RBF kernel enforces a strong prior for smoothness in this learned space Z: if two points $h_i(\psi(x))$ and $h_i(\psi(x'))$ are close in Euclidean distance, their corresponding function values $f(x)$ and $f(x')$ are modeled to be highly correlated.

Therefore, the space of priors that can be fruitfully expressed is the set of all functions that can be simplified to a stationary smoothness assumption through a continuous mapping. This architecture is naturally suited for problems where the data generating function $f_{dg}(x) : \mathbb{R}^D \to \mathbb{R}$ exhibits a specific structure. The hypothesis is that while $f_{dg}$ may be complex in the high-dimensional input space $\mathcal{X}$, its structure is fundamentally governed by a smaller number of task-specific continuous latent variables, $v$. More formally, we assume there exists a low-dimensional manifold $\mathcal{M}$ parameterized by latent coordinates $v \in \mathcal{V}$ (where dim($\mathcal{V}$)<dim($\mathcal{X}$)) via a map $g : \mathcal{V} \to \mathcal{M} \subset \mathcal{X}$. The core hypothesis is that the true function can be expressed as the composition of a "simple" function $f_{simple} : \mathcal{Z} \to \mathcal{R}$ and the inverse of the parameterization, $g^{-1} : f_{dg}(x) \approx f_{simple}(g^{-1}(x))$ for $x \in \mathcal{M}$

The role of the deep network and linear head is to help approximate this $g^{-1}$, allowing *our* $f_{simple}$ (an RBF Gaussian Process) to be a good prior for the data generating function.

This is closely related the *manifold hypothesis* in machine learning (Bengio et al., 2013; Calandra et al., 2016), and is extraordinarily general across natural science datasets. **We provide numerous examples** in Table S2.

### S.C.2    GENERALITY VIA A MODULAR META-LEARNING PIPELINE

The generality of our work stems not only from the deep kernel architecture but from the underlying meta-learning pipeline itself. The framework's components are modular and can be adapted to problems of varying complexity and data modality.

**Adapting the Feature Extractor to Data Modality:**    While we employ a CNN for image data, the feature extractor $\psi$ is a pluggable module. For problems with different data structures, it could be replaced with a more suitable architecture, such as a Graph Neural Network for molecular data or a Transformer for sequence data, without changing the overall framework.

**Adapting the Kernel to Problem Complexity:**    The choice of a deep kernel is motivated by our complex, high-dimensional problem. For simpler scientific problems defined on low-dimensional inputs (e.g., modeling 1D orientation tuning curves), a deep feature extractor would be unnecessary. In such cases, our framework could be instantiated by meta-learning the hyperparameters of an expressive classical kernel, such as the **Spectral Mixture Kernel** (Wilson & Adams, 2013), directly from theory-generated tasks. This demonstrates that the core principle—meta-learning a prior from a theory—is broadly applicable across different model complexities.

| System to Model | Input $\mathcal{X}$ | Output $y$ | Latent Variable $\mathcal{V}$ | (Normative) Theory |
|---|---|---|---|---|
| Primary Visual Cortex (V1) | Images | Single Neuron firing rates | Stimulus features (orientation, spatial frequency, motion) | Sparse Coding (Olshausen & Field, 1996) |
| Primary Motor Cortex (M1) | Neural Population Activity | Limb Velocity vector | Intended Limb Kinematics (Position, Velocity, Acceleration) | Optimal Feedback Control (Todorov & Jordan, 2002; Vargas et al., 2024) |
| Primary Auditory Cortex (A1) | A sound's time-frequency spectrogram | Single Neuron firing rate | Spectro-temporal features like pitch, timbre | Efficient Coding of Natural Sounds (Lewicki, 2002) |
| Hippocampal Place Cells | Multi-Sensory inputs | Place Cell firing | The animal's 2D spatial position, proximity to abstract goals (El-Gaby et al., 2024), and head direction. | Successor Representations (Momennejad et al., 2017) |
| Prefrontal Cortex (PFC) | Stimuli and Reward history | PFC population activity | Abstract Task Variables ("belief state", "decision rule") | Bayesian Decision theory (Körding & Wolpert, 2006), Drift Diffusion models |
| Protein Science | Amino Acid Sequence | A functional property (e.g., catalytic efficiency, stability) | Geometric or electrostatic properties of the protein's active site | Molecular Dynamics simulations |
| Molecular Property prediction | Small Molecule features (Connectivity Fingerprint) | reactivity (EC50) | pharmacophores or structural motifs | DFT or QSAR predictions |
| Developmental Biology | Vector of local morphogen concentrations | Cell-fate gene expression level | Cell's physical position along a developmental axis | Reaction-Diffusion Models (Zagorski et al., 2017) |

Table S2: A selection of scientific systems where the core hypothesis of our framework may be applicable.

## S.D  BIOLOGICAL DATA PREPROCESSING

### S.D.1  DATASET SOURCE AND INITIAL PROCESSING BY QIU ET AL.

Our dataset was sourced from the published work of Qiu et al (Qiu et al., 2023b;a), focusing specifically on the responses of 86 mouse ventral ganglion cells to a 30Hz natural movie stimulus with Green and Ultraviolet (UV) channels, as described in our main text. To reiterate, each input to our regression model is derived from the downsampled image frames (36×32 pixels) presented at 30Hz to the retinal cells. The outputs we predicted are the Calcium fluorescence responses measured for each neuron, acquired via two-photon microscopy at a frame rate of 7.8125 Hz. These fluorescence signals serve as proxies for neuronal firing activity, as calcium indicators reflect intracellular calcium concentration changes associated with cells firing action potentials. While we applied our own our processing steps (temporal flattening via a canonical filter derived from an LN model, hierarchical clustering to select unique representative images, selection of the UV channel, and z-scoring), Qiu et al. performed crucial initial preprocessing on the data, most important of which we describe here- but refer readers to their publication for full details.

**Cell Selection via Quality Index (QI)**   The dataset includes responses from cells selected based on their response reliability. Qiu et al. calculated a Quality Index (QI), ranging from 0 to 1, for each cell based on the consistency of its responses across repeated presentations of specific test sequences. The QI was calculated as:
$$QI = \frac{\text{Var}_t x \left( E_r[C] \right)}{E_r \left( \text{Var}_t[C] \right)}$$
where $C$ represents the response matrix with dimensions time samples ($t$) by repetitions ($r$), $E_r[\cdot]$ denotes the expectation (average) across repetitions, and $\text{Var}_t(\cdot)$ denotes the variance across time samples. A higher QI indicates a more reliable response with a higher signal-to-noise ratio. For inclusion in the final dataset used for modeling, neurons responding to the natural movie stimulus had to meet the criterion $QI_{\text{noise}} > 0.25$, where $QI_{\text{noise}}$ was determined using responses to separate noise stimulus test sequences also presented during the experiments.

**Stimulus and Response Handling**   Importantly for evaluation, the 750 test and validation fluorescence responses provided in the dataset are not raw signals but represent an average across 3 repetitions of specific held-out input sequences from the natural movie stimulus. This averaging serves to increase the signal-to-noise ratio of these benchmark traces. This makes the original train/test+val split non-interchangeable.

For exhaustive details on the experimental procedures (including animal handling, tissue preparation, recording setup) and the complete preprocessing pipelines applied across all stimulus conditions in the original study, we refer the reader to the source publications (Qiu et al., 2023b;a). Our subsequent processing steps applied for this work are detailed below.

### S.D.2  TEMPORAL FILTER EXTRACTION VIA LN MODEL

To convert the regression problem from movie inputs to image inputs, we temporally flatten the movies. To achieve this, we first fit a spatiotemporal Linear-Nonlinear (LN) model to predict the 16200 neural responses $\in \mathbb{R}^{16200 \times 86}$ from 16200 movie inputs $\in \mathbb{R}^{16200 \times 1152 \times 50}$. This LN model consists of spatial filter parameters $W \in \mathbb{R}^{86 \times 1152}$ and Temporal filter parameters $\tau \in \mathbb{R}^{86 \times 50}$. Among temporal filters we observed classic biphasic shapes (dip-then-peak or vice versa). We then "flipped" some filters by multiplying by -1 in order to align their biphasic structures and allow for averaging, and then averaged across neurons to obtain a single "canonical" filter $F \in \mathbb{R}^{50}$, spanning 50 frames. The choice of which filters needed flipping was made by thresholding at the peak value. Each movie frame $x_\tau \in \mathbb{R}^{36 \times 32}$ was then projected onto this canonical temporal filter, effectively collapsing the time dimension into a single static "image", $\tilde{x} = \sum_{\tau=1}^{50} F_\tau x_\tau$. We plot all 86 time filters identified this way (Fig. S4) alongside the canonical average filter.

### S.D.3  FILTERING NON REDUNDANT IMAGES

To derive a representative subset from the 16,200 images in our dataset, we applied a data-driven hierarchical clustering approach. We first computed 1-NN distances (excluding self-matches) in the

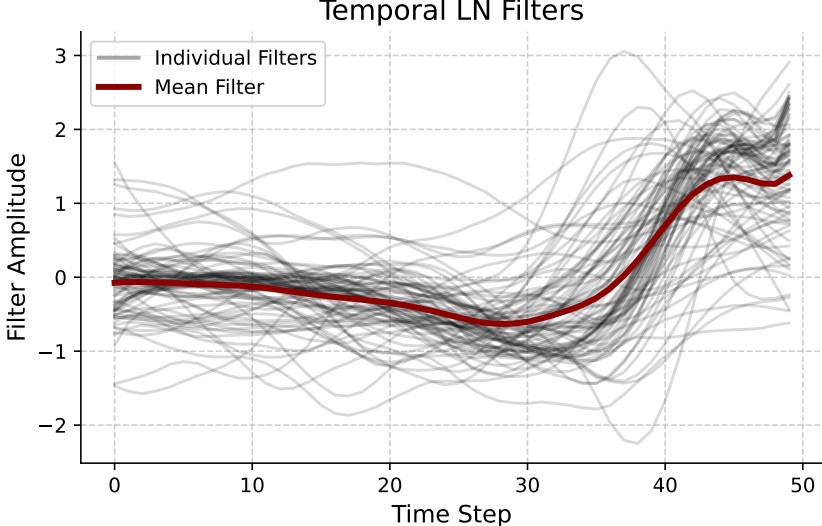

Figure S4: **Time Filters of LN model on Biological Data** Individual filters shown are post-flipping.

feature space (shaped $16200 \times 1152$) using Scikit-Learn's `NearestNeighbors` with Euclidean distance. These distances were sorted, and a clustering threshold was selected based on the largest gap between successive values—a heuristic for natural cluster separation. We then performed agglomerative clustering with average linkage and this threshold (with `n_clusters=None`), resulting in 1,452 clusters. A single exemplar (the first image) from each cluster formed our representative subset.

## S.E  SYNTHETIC META-TRAIN DATASET GENERATION

This section outlines the generation of the synthetic dataset based on our chosen efficient coding theory.

### S.E.1  NORMATIVE MODEL (EFFICIENT CODING AUTOENCODER)

**Architecture of Efficient Coding Autoencoder**  Table S3 shows the full architecture used.

| Layer | Details | Output Shape | Activation |
|---|---|---|---|
| Input | - | B, 1, 28, 28 | - |
| Conv | 9×9, padding=4 | B, 16, 28, 28 | ReLU |
| Flatten | - | B, 12544 | - |
| Linear1 | $12544 \rightarrow 300$ | B, 300 | - |
| Gaussian Noise | $\sigma = 2.0$ (train only) | B, 300 | - |
| Bottleneck | - | B, 300 | ReLU |
| Linear2 | $300 \rightarrow 12544$ | B, 12544 | ReLU |
| Unflatten | - | B, 16, 28, 28 | - |
| ConvTranspose | 9×9, padding=4 | B, 1, 28, 28 | Tanh |
| Output | - | B, 1, 28, 28 | - |

Table S3: **Normative Autoencoder Architecture**

**Training Procedure**  The autoencoder model, adapted from work by Ocko et al. and Qiu et al. (Ocko et al., 2018; Qiu et al., 2023b), was trained to learn an efficient code for natural images. The core idea of efficient coding is that sensory systems aim to represent naturalistic stimuli faithfully while minimizing metabolic costs. Our autoencoder operationalizes this by optimizing the loss function $\mathcal{L}_{\text{EC}}$; defined as:

$$\mathcal{L}_{\text{EC}} = \text{MSE}(X, \hat{X}) + \beta ||A_{\text{bottleneck}}||_1 + \alpha \left( ||W_{\text{conv}}||_2^2 + ||W_{\text{tconv}}||_2^2 \right) \tag{S1}$$

where $\text{MSE}(X, \hat{X}) = \frac{1}{N} \sum_{j=1}^{N} ||X_j - \hat{X}_j||_2^2$ is the mean squared error ensuring fidelity between original ($X_j$) and reconstructed ($\hat{X}_j$) images over a batch of $N = 200$ samples. The second term applies an L1 penalty to the bottleneck layer activations $A_{\text{bottleneck}}$ with weight $\beta = 1 \times 10^{-5}$ to promote sparse codes. The third term imposes an L2 penalty on the weights of convolutional ($W_{\text{conv}}$) and transposed convolutional ($W_{\text{tconv}}$) filters with weight $\alpha = 5 \times 10^{-5}$ to encourage smooth filters. Optimization was performed over 200 epochs using the Adam optimizer (learning rate $1 \times 10^{-5}$). We trained on 60000 patches sampled from the public Van-Hateren dataset of natural images (Van Hateren & van der Schaaf, 1998).

Consistent with efficient coding frameworks that account for noise, additive Gaussian noise (std 2) was applied to the input images during training, simulating sensory noise and acting as a regularizer. While Ocko et al. (Ocko et al., 2018) primarily used noise and sparsity penalties without an explicit narrow bottleneck in their model, our implementation imposed a fixed bottleneck size. To ensure we were meta-training on a high-fidelity representation of the theory, we **performed a small hyperparameter search** over the normative model's architecture and selected a 300-unit bottleneck, along with the other hyperparameter values. This was a pragmatic choice that reliably produced the canonical center-surround receptive fields predicted by efficient coding. Our chosen hyperparameters ($\alpha$, $\beta$, input noise, bottleneck size), represent one point in a large trade-off space between reconstruction accuracy, code sparsity, and filter smoothness, and were chosen to ensure a robust and faithful instantiation of the theory for our subsequent meta-learning. Varying these parameters would likely yield different efficient coding solutions, resulting in different filter properties (e.g. center/surround relative and absolute size) as the model adapts to different balances of these competing objectives under varying architectural and noise constraints.

**Example Receptive Fields**  To validate the features learned by the hidden units in the autoencoder's bottleneck layer, we visualized the preferred input pattern associated with each unit. This was achieved by activating each of the 300 bottleneck units individually (setting its value to 1 while others were 0) and passing this sparse representation through the trained decoder network. The resulting decoded output image represents the spatial pattern that maximally drives (or is reconstructed by) that specific bottleneck unit (Fig. S5)

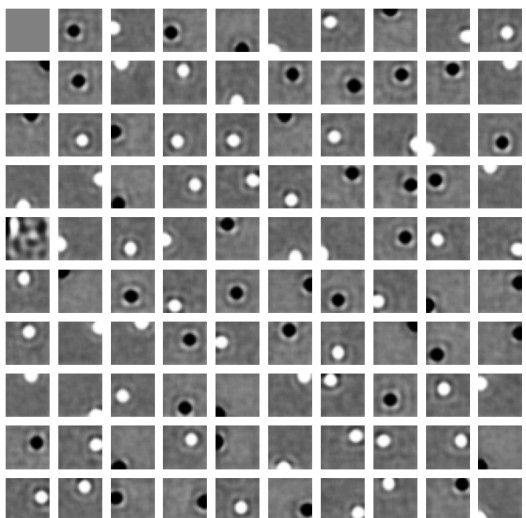

Figure S5: **Normative Autoencoder Receptive Fields**

### S.E.2    DATA AUGMENTATIONS PERFORMED TO EXTEND THE META-TRAIN SET

To generate the synthetic "tasks", we extract spatial receptive fields of the "optimal" bottleneck neurons of our efficient coding autoencoder by fitting a spatial Linear-Nonlinear (LN) model to the

activations of the autoencoder bottleneck. This yields 300 center-surround filters $f_i$ . Each filter then defines one synthetic task obtained by taking its dot product with the image stimuli $y_{i,k} = f_i^T x_k$ where $x_k$ is the $k'th$ image. We expand this meta-train set by choosing 20 archetypal un-obfuscated center surround filters (by visual inspection), **rescaling their size** (randomly ranging from 0.8x to 1.2x) and **jittering** the location of their 'centers' , allowing us to increase the size of the synthetic meta-train set $N_{\mathcal{T}}$. We created 4000 such synthetic tasks, but ultimately used only $N_{\mathcal{T}} = 490$ selected randomly, as we found empirically that this number of tasks saturated the meta-training performance.

**Rationale behind Jittering and Scaling.** Our Jittering and Scaling augmentations are not arbitrary, but rather a principled computationally efficient way to generate new tasks consistent with the same normative theory. Our meta learning framework treats each neuron as an individual "task", meaning our framework learns a prior for **single-neuron** responses. Our meta-train set is therefore composed of many separate individual examples of "good" single neuron solutions consistent with the theory which is itself posed at a population level. On a single neuron level, efficient coding theories predict a *type* of solution (a center-surround structure), not one single filter at one specific pixel location. Therefore, **a filter jittered to a different location is an equally valid example of a high-performing solution under the theory of Ocko et al**: it would have effectively the same utility level. This is in stark contrast to population level efficient coding theories, which may precisely specify the relative positions, overlap levels, and tilings of a population of efficient coding neurons- so if we were learning a population-level prior rather than a single neuron level prior, the jittering would be problematic as it would be deleting that information.

The **spatial scale of receptive fields**, on the other hand, is a known consequence of architectural capacity and regularization in efficient coding models. Ocko et al, for instance, have analytically connected the size of the emergent receptive fields to the "stride length" of the convolutional filters-driven by the need to prevent aliasing. Other seminal works, such as that of Karklin and Simoncelli (Karklin & Simoncelli, 2011), have demonstrated that the "optimal" efficient receptive field size is a function of the input noise level. Because the noise is applied i.i.d to each pixel, a high level of noise primarily affects the Signal-to-noise ratio of high-frequency components of the image's power spectrum. Therefore, in a high-noise environment, the optimal solution is to average over a larger spatial area to filter out the noise. This corresponds to a larger receptive field. On the other hand, given low noise, the high frequency features are more reliable, and therefore valuable to capture with spatially smaller receptive fields.

There is not a single efficient coding theory predicting a single "optimally sized" canonical receptive field. The **optimal size is conditional** on the stimulus statistics and architectural constraints. In fact, biological retinal ganglion cells are known to adapt the sizes of their functional receptive fields via inhibitory plasticity precisely in response to these changing environmental statistics (Smirnakis et al., 1997). Therefore, our choice to spatially scale the sizes of receptive fields in a reasonable range is a principled, computationally efficient way to convert our Efficient Coding solutions trained at a single fixed noise level and architecture into a family of solutions consistent with a larger range of these constraints- which are in turn all consistent with the theory- being optimal solutions corresponding to different hyperparameter values.

If we had known the precise noise levels experienced by the photoreceptor cells of the *ex vivo* biological neurons, we may try to reverse engineer a single noise level in our model consistent with their true underlying noise level and test the predictions of the theory at that level of specificity. However, since that is far beyond our scope, **scaling the receptive fields the most principled thing to do**: resulting in a set of meta-train tasks representative of a wide range of these hyperparameter values which we as scientists do not know the ground truth value of.

Nonetheless, we provide an **Ablation study on the Scaling data augmentation**, demonstrating that our model's accuracy is, if anything, reduced by the preprocessing across most data sizes.

| N | 4 | 8 | 16 | 32 | 64 |
|---|---|---|---|---|---|
| Theory Informed Kernel | $0.015 \pm 0.015$ | $0.072 \pm 0.019$ | $0.135 \pm 0.017$ | $0.182 \pm 0.019$ | $0.217 \pm 0.014$ |
| Ablation: No Scaling | $0.021 \pm 0.015$ | $0.079 \pm 0.022$ | $0.154 \pm 0.020$ | $0.192 \pm 0.022$ | $0.231 \pm 0.010$ |

| N | 128 | 256 | 512 | 1024 | 1422 |
|---|---|---|---|---|---|
| Theory Informed Kernel | $0.250 \pm 0.005$ | $0.287 \pm 0.004$ | $0.307 \pm 0.005$ | $0.323 \pm 0.004$ | $0.325 \pm 0.002$ |
| Ablation: No Scaling | $0.256 \pm 0.008$ | $0.289 \pm 0.007$ | $0.305 \pm 0.005$ | $0.314 \pm 0.002$ | $0.324 \pm 0.001$ |

## S.F   META-LEARNING ALGORITHM AND TRAINING DETAILS

### S.F.1   MODEL ARCHITECTURE

As a general principle, tried to minimize the flexibility of our task-adaptive components, as doing so in conjunction with bilevel optimization may reduce pressure on the meta learned feature extractor to learn during meta training- since the task adaptive components would be flexible enough to capture the requisite structure on their own.

**Feature Extractor Architecture**   Table S4 shows our feature extractor architecture.

| Layer | Details | Output Shape | Activation |
|---|---|---|---|
| Input | - | B, 1, 36, 32 | - |
| Conv1 | 3×3, padding=1 | B, 32, 36, 32 | GeLU |
| Conv2 | 3×3, padding=1 | B, 64, 36, 32 | GeLU |
| MaxPool | 2×2 | B, 64, 18, 16 | - |
| Conv3 | 3×3, padding=1 | B, 128, 18, 16 | GeLU |
| Conv4 | 3×3, padding=1 | B, 128, 18, 16 | GeLU |
| Flatten | - | B, 36864 | - |
| Linear1 | $36864 \rightarrow 256$ | B, 256 | GeLU |
| Linear2 | $256 \rightarrow 256$ | B, 256 | - |

Table S4: **CNN feature extractor ($\phi$) architecture**

**Linear Heads**   All linear heads are `torch.nn.Linear` instances, with 256 input dims and 128 output dims and L1 regularization (=0.01) to encourage generalization (Fumero et al., 2023), also following the recommendation from (Van Amersfoort et al., 2021) to not overly shrink the latent dimensionalities.

**Gaussian Process Layer**   For all of our GP layers, we used a GPyTorch Exact GP with an RBF Kernel $k(x, x'; \theta_{\text{gp}}) = \sigma_f \exp(-\frac{\|x-x'\|^2}{2l^2})$. We set all mean functions to `ZeroMean()`, and used a Gaussian Lengthscale Prior with a variance of 0.01 centered around the initialization lengthscale value. For initialization, we followed the heuristic "median lengthscale init" reported in previous work (e.g. (Van Amersfoort et al., 2021)) where the lengthscale is initialized to be the median distance between the embedded data points. Importantly, we did *not* keep re-computing the median init as we cycled through task batches, instead saving the first computed median and re-initializing with that value. Experimental validation demonstrated the importance of this fixed median initialization and narrow lengthscale hyper-prior for avoiding feature collapse (Sec. S.F.4)

### S.F.2   META-TRAINING PROCEDURE

Table S5: Hyperparameters for the Optimization Procedure (Algorithm 1)

| Hyperparameter | Symbol | Description | Value |
|---|---|---|---|
| Number of Epochs | $E$ | Total number of meta-training epochs. | 7 |
| Batch Size | $B$ | Number of tasks per meta-batch. | 10 |
| Inner Adaptation Steps | $n_{\text{inner steps}}$ | Number of gradient steps for task adaptation. | 17 |
| Inner Learning Rate (Linear) | $\alpha_{\text{linear}}$ | Learning rate for linear head parameters in inner loop. | 0.0004 |
| Inner Learning Rate (GP) | $\alpha_{\text{GP}}$ | Learning rate for GP hyperparameters in inner loop. | 0.002 |
| Outer Meta-Update Steps | $n_{\text{outer steps}}$ | Number of outer loop optimization gradient steps | 3 |
| Outer Learning Rate | $\alpha_{\text{outer}}$ | Learning rate for meta-parameters ($\theta_{\text{meta}}$) in outer loop. | 0.0004 |

Our training procedure (Alg. 1) is similar to that of the "Adaptive Deep Kernel Fitting via the Implicit Function Theorem" (ADKF-IFT) method by Chen et al (Chen et al., 2023). We employ Bi-level

---

**Algorithm 1** Procedure for Fitting Adaptive Deep Kernel Model

---

1: **Define:** Task-adaptive parameters $\theta_{\text{adapt}}$ (i.e. $\theta_{\text{gp},i}$ and $W_i$ from $h_i$)
2: **Define:** Meta-learned parameters $\theta_{\text{meta}}$ (i.e. $\theta_{\text{nn}}$ from $\phi$)
3: **Input:** Meta training set

$$\mathcal{T} = \left\{ \mathcal{T}_i = \left( \mathcal{D}_i^{\text{support}}, \mathcal{D}_i^{\text{query}} \right) \right\}_{i=1}^{N_T},$$

where for each task $\mathcal{T}_i$:

$$\mathcal{D}_i^{\text{support}} = \{(x_{i,k}^{\text{support}}, y_{i,k}^{\text{support}})\}_{k=1}^{n_i^{\text{support}}} \quad \text{and} \quad \mathcal{D}_i^{\text{query}} = \{(x_{i,k}^{\text{query}}, y_{i,k}^{\text{query}})\}_{k=1}^{n_i^{\text{query}}}.$$

4: Shuffle $\mathcal{T}$ and partition into batches
5: **for** epoch $= 1$ **to** $E$ **do**
6:     **for** batch $\in \mathcal{T}$ **do**
7:         **for** each task $\mathcal{T}_i = \left( \mathcal{D}_i^{\text{support}}, \mathcal{D}_i^{\text{query}} \right)$ in the batch **do**
8:             Reinitialize task-adaptive parameters $\theta_{\text{adapt},i}$
9:             **for** $t = 1$ **to** $n_{\text{inner steps}}$ **do**
10:                 Update task-specific parameters:

$$\theta_{\text{adapt},i} \leftarrow \theta_{\text{adapt},i} - \alpha_{\text{inner}} \nabla_{\theta_{\text{adapt}}} \log P\left( y_i^{\text{support}} \mid x_i^{\text{support}}, \theta_{\text{adapt},i}, \theta_{\text{meta}} \right),$$

                where $\alpha_{\text{inner}} = (\alpha_{\text{linear}}, \alpha_{\text{GP}})$ denotes the learning rates for the linear head and GP hyperparameters, respectively.
11:             **end for**
12:             Set $\theta_{\text{adapt},i}^* \leftarrow \theta_{\text{adapt},i}$.
13:         **end for**
14:         **for** $j = 1$ **to** $n_{\text{outer steps}}$ **do**
15:             Update the meta-parameters:

$$\theta_{\text{meta}} \leftarrow \theta_{\text{meta}} - \alpha_{\text{outer}} \nabla_{\theta_{\text{meta}}} \mathbb{E}_{i \in \text{batch}} \log P\left( y_i^{\text{query}} \mid x_i^{\text{query}}, \mathcal{D}_i^{\text{support}}, \theta_{\text{adapt},i}^*, \theta_{\text{meta}} \right),$$

            where $\alpha_{\text{outer}}$ is the meta-learning rate.
16:         **end for**
17:     **end for**
18: **end for**

---

optimization: with disjoint task-adaptive and meta-learned modules trained separately, on different learning objectives. Like Chen et al, we use the standard GP marginal log likelihood (Alg. 1,Eq. 1, main paper) for our "inner" training loop on task-adaptive parameters and the log-probability of the query data for the "outer" training loop on meta-learned parameters (to encourage generalization (Lotfi et al., 2022)). Unlike Chen et al, we do not use implicit differentiation to enrich the outer level gradient, as this would not be computationally feasible with our significantly higher number of task-adaptive parameters. Future work may explore the likely benefits of employing a scheme to approximate these implicit gradient terms (Geng et al., 2021). We also make adjustments to the training process to avoid feature collapse, discussed in S.F.4

We trained on batches of 10 (synthetic) tasks at a time, each task containing 1452 labeled input/output training data, randomly sampling 49 such batches, resulting in a total of 490 tasks in our meta train set. We trained for 7 epochs in total. At the start of every epoch, we ran the full adaptation procedure on three meta sets, using 100 support points each. One on Synthetic data from the meta train set. One on Synthetic data unseen during meta training but from the same distribution. One on our Biological data, with all evaluations computed on the validation data only, using a subset of tasks. We employed early stopping based on held-out biological data Pearson correlation (Fig. S6). During our training process, we clipped the gradients on the base feature extractor to have a maximum norm of 1, and we trained the first epoch with 1/10th the learning rate. As shown in Alg. 1, we optimized the feature extractor parameters $\theta_{nn}$ on the log-probability of the query data (outer level) in the meta-train set and the linear heads / GP layer parameters ($W_i$ and $\theta_{\text{gp},i}$) on the MLL of the support set (inner level). For synthetic meta train sets, we manually set the GP noise level close to zero to avoid feature collapse-preventing the log-prob objective from hiding collapsing epistemic uncertainty behind high aleatoric uncertainty (as discussed in S.F.4). Our experiments indicated that utilizing a smaller proportion of data points for the support set, with a correspondingly larger proportion for the query set, fostered more effective representation learning in the feature extractor. In our case, out of 1452 possible points to be divided into both for each task, we used $5\%$ of them for our support set, and $95\%$ for our Query set, forcing the model to predict more unseen points using fewer conditioned values, and re-sampling the exact support/query split every epoch. We also adhered to the `torch.double()` dtype to avoid numerical errors from the low noise level.

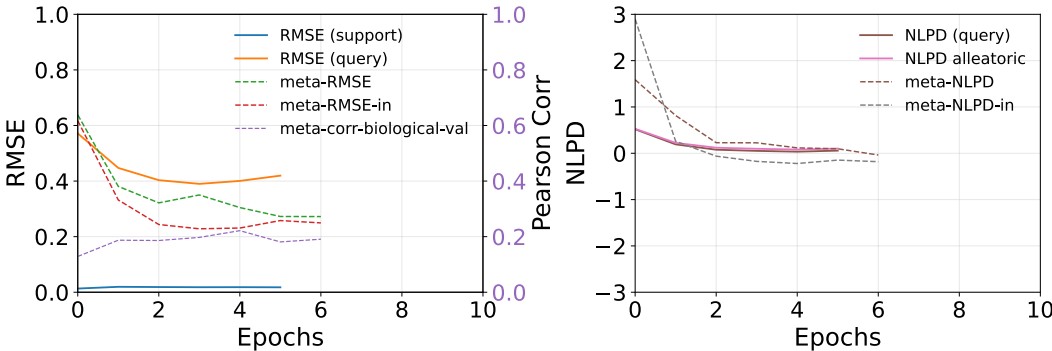

Figure S6: **Training Stats**,Left Plot, Left Axis corresponds to mean RMSE (averaged over tasks), right axis corresponds to pearson correlation averaged over tasks (biological). Right plot depicts NLPD values, with NLPD(query) depicting epistemic NLPD and NLPD alleatoric including the noise term (which here is set to be very low). Dotted lines signify post-adaptation tests of the learned feature extractor at the start of each epoch. Solid lines are stats accumulated during training. The dotted lines are evaluated at the start of every epoch, while the solid lines are evaluated at the end of every epoch, which is why there is one extra dotted line point. Meta RMSE and Meta NLPD correspond to unseen tasks drawn from the same distribution, and meta-rmse-in and meta nlpd in correspond to tasks from the meta train set. To be clear, we early-stopped using the dotted purple line, which reflects what a scientist using our method should/would do.

### S.F.3  TASK ADAPTATION PROCEDURE

For each meta-test task (i.e., each biological neuron), we perform task adaptation to fit the model to the available biological data (the support set for that task). This involves optimizing only the task-specific parameters while keeping the meta-learned feature extractor weights frozen. The parameters adapted for each task $i$ include the weights of its linear head ($W_i$) and the hyperparameters of its Gaussian Process (GP) layer $\theta_{gp,i} = (\sigma_{f,i}, \ell_i, \sigma_{\eta,i})$, encompassing the GP output scale, length scale, and likelihood noise variance, respectively.

Adaptation is performed by maximizing the marginal likelihood of the support data for that task using the Adam optimizer (Kingma & Ba, 2014) for 300 epochs. We used a base learning rate of $4 \times 10^{-3}$ for the GP hyperparameters and Adam betas of $(0.99, 0.999)$. The learning rate for the linear head weights $W_i$ was scaled down by a factor of $10^{-2}$ relative to the base rate (i.e., $4 \times 10^{-5}$). Additionally, an L1 sparsity penalty with a coefficient of $10^{-2}$ was applied to the linear head weights during adaptation. For the GP hyperparameters, the length-scale prior was kept narrow (Gaussian with variance 0.01, except for the RBF-only case) and its mean was initialized using the median heuristic applied to the pairwise distances of the features $d(x, x') = ||h_i(\phi(x)) - h_i(\phi(x'))||_2$ from the frozen extractor and freshly initialized heads. For experiments involving synthetic meta-test data, the GP likelihood noise $\sigma_{\eta,i}$ was initialized at $10^{-4}$- reflecting known ground truth. For biological data, the standard GPyTorch initialization was used.

Crucially, these same adaptation settings (optimizer, learning rates, epochs, priors, initializations) were used consistently across our main 'Informed' model and all ablation conditions. Given the significant computational cost required for extensive hyperparameter tuning for each model variant across dozens of tasks and multiple data subsets, we adopted these fixed, reasonable settings based on preliminary experiments. The consistent application ensures a fair comparison between the conditions, and the sensible trends observed in the results (e.g., Fig. S1) suggest that these settings, while perhaps not perfectly optimal for every single task/model pairing, were adequate and did not unduly handicap any specific model.

### S.F.4  DESIGN CHOICES MITIGATING FEATURE COLLAPSE IN OUR DEEP KERNEL MODEL

Although literature such as (Chen et al., 2023) suggests that meta-learning across diverse tasks might inherently mitigate feature collapse, our experiments indicated this was not guaranteed for our use

case. However, we successfully avoided feature collapse in our model, attributing this primarily to three specific design choices in our training methodology:

1. **Narrow Hyperprior on GP Lengthscales:** We imposed a tight prior distribution on the Gaussian Process (GP) lengthscale parameters ($\ell_i$), severely restricting their ability to adapt during the inner-loop optimization for each task.

2. **Fixed GP Lengthscale Initialization:** We did not re-initialize the GP lengthscales using the median heuristic at the start of each task batch adaptation, a technique commonly used to set reasonable initial values (e.g., Chen et al., 2023; Van Amersfoort et al., 2021). We used the median heuristic to compute the lengthscale once, at the start of the first task batch, and then used that lengthscale throughout the other taskbatches.

3. **Epistemic Log Probability Objective:** Our outer-loop meta-training objective optimized the feature extractor ($\phi$) based on the log probability of the query data ($y_q$) conditioned on the support data (($x_s, y_s$)), often referred to as the negative log predictive density (NLPD). Unlike existing training approaches, however, we explicitly used the log probability of the data excluding the likelihood noise. This focuses purely on the epistemic uncertainty captured by the GP, unlike objectives combining epistemic and aleatoric uncertainty (e.g., Chen et al., 2023; Lotfi et al., 2022) or optimizing the log marginal likelihood (LML) directly (e.g., Patacchiola et al., 2020).

Our preliminary understanding of the mechanisms by which these choices prevent collapse follows:

**Choices 1 and 2 (Lengthscale Restriction):** We observed empirically that allowing the GP lengthscale ($l_i$) to be freely optimized within each task's inner loop (as is typical in GP regression and done for task-adaptation in (Chen et al., 2023)) or frequently re-initializing it via the median heuristic, created a detrimental positive feedback loop during meta-training. Shorter lengthscales appeared to incentivize the deep feature extractor ($\phi$) to map inputs closer together in feature space. This proximity, in turn, further encouraged shorter lengthscales via the inner-loop LML optimization, ultimately leading to collapsed feature representations where distinct inputs map to nearly identical points. By fixing the lengthscale via a narrow prior and consistent initialization, we broke this feedback loop. The potential loss of flexibility from a fixed lengthscale was compensated by the adaptability of the meta-learned feature extractor ($\phi$) and the task-specific linear heads ($h_i$), which could still arrange the features appropriately for the given lengthscale.

**Choice 3 (Epistemic Log Probability Objective):** This choice appears to counteract feature collapse through two mechanisms:

**Avoiding Prior Log-Determinant Minimization:** Ober et al. (Ober et al., 2021) identified a key mechanism driving feature collapse in DKL models trained via LML maximization. The LML objective can be decomposed into a data fit term and a complexity penalty term:

$$\log p(y) = \underbrace{-\frac{1}{2}y^T(K + \sigma_\eta^2 I_N)^{-1}y}_{\text{Data Fit Term}} \underbrace{-\frac{1}{2}\log|K + \sigma_\eta^2 I_N|}_{\text{Complexity Penalty Term}} - \frac{N}{2}\log(2\pi)$$

Crucially, Ober et al. prove (in their Proposition 1) that for kernels with a learnable output scale ($\sigma_f$), the data fit term becomes a constant ($-N/2$) at the optimum. Consequently, maximizing the LML with highly flexible feature extractors becomes dominated by minimizing the complexity penalty, specifically the log determinant term $\log|K + \sigma_\eta^2 I_N|$. The model achieves this by learning feature mappings ($\phi$) that make the kernel matrix $K$ (evaluated on the learned features) as close to singular as possible, which corresponds to collapsing the features.

Our objective, however, focuses on the log probability of query data $y_q$ given support data ($X_s, y_s$), which depends on the *posterior* predictive distribution for the latent function values $f_q$ at query inputs $X_q$. This distribution is Gaussian, $p(f_q|X_q, X_s, y_s) = \mathcal{N}(f_q|\mu_{q|s}, \Sigma_{q|s})$. Let $X_s = \{x_{s,1}, \ldots, x_{s,n_s}\}$ be the set of $n_s$ support inputs and $X_q = \{x_{q,1}, \ldots, x_{q,n_q}\}$ be the set of $n_q$ query inputs. The kernel function, parameterized by the feature extractor $\phi$, is $k_\phi(\cdot, \cdot)$. We define the following kernel matrices:

- $K_{ss} = [k_\phi(x_{s,i}, x_{s,j})]_{i,j=1}^{n_s}$, the $n_s \times n_s$ kernel matrix between support points.

- $K_{qq} = [k_\phi(x_{q,i}, x_{q,j})]_{i,j=1}^{n_q}$, the $n_q \times n_q$ kernel matrix between query points.

- $K_{qs} = [k_\phi(x_{q,i}, x_{s,j})]_{i=1,j=1}^{n_q,n_s}$, the $n_q \times n_s$ kernel matrix between query and support points.

- $K_{sq} = K_{qs}^T$, the $n_s \times n_q$ kernel matrix between support and query points.

The posterior predictive covariance is then given by $\Sigma_{q|s} = K_{qq} - K_{qs}(K_{ss} + \sigma_n^2 I_{N_s})^{-1} K_{sq}$, where $\sigma_n^2$ is the observational noise variance and $I_{n_s}$ is the $n_s \times n_s$ identity matrix. The relevant term in our objective related to model complexity is $-\frac{1}{2} \log |\Sigma_{q|s}|$. By properties of determinants of positive definite block matrices (specifically, the joint prior covariance matrix over $f_s$ and $f_q$), it follows that $|\Sigma_{q|s}| \leq |K_{qq}|$. Therefore, $\log |\Sigma_{q|s}| \leq \log |K_{qq}|$, and minimizing $\log |\Sigma_{q|s}|$ as part of our objective is a different optimization target than directly minimizing $\log |K_{qq}|$. We hypothesize that this difference mitigates some of the detrimental effect of minimizing the log determinant of the *prior* covariance (i.e., $\log |K_{qq}|$ or terms in $\log |K_{ss} + \sigma_n^2 I_{n_s}|$), reducing our susceptibility to the failure mode studied by Ober et al. Notably, the outer meta-learning loop objective used by (Chen et al., 2023) is also based on the query set predictive posterior, suggesting they might also have benefited from this effect.

**Explicit Epistemic Uncertainty Optimization:** A more speculative reason relates to the nature of the objective. Feature collapse in deep kernel GPs affects the uncertainty of their latent function values. Therefore, optimizing for an objective that explicitly depends on the latent function uncertainty (i.e., epistemic uncertainty via $\Sigma_{q|s}$) may have encouraged the formation of features that avoid collapse. Specifically, overconfident latent function predictions associated with feature collapse (i.e., near-singular $\Sigma_{q|s}$) would be heavily penalized by the data-fit component of the loss function (which involves $\Sigma_{q|s}^{-1}$), unless predictions were unrealistically perfect. This leads to a poor overall log probability. If we had trained on an objective incorporating observed (aleatoric) noise, a sufficiently large learned noise variance ($\sigma_{\text{aleatoric}}^2$) could stabilize the predictive covariance (i.e., in terms like $\Sigma_{q|s} + \sigma_{\text{aleatoric}}^2 I$), thereby masking the collapsed epistemic uncertainty $\Sigma_{q|s}$ and reducing the direct pressure on the feature extractor to maintain well-calibrated epistemic uncertainty.

## S.G IMPLEMENTATION DETAILS AND RUNTIMES

### S.G.1 SOFTWARE AND HARDWARE

All models were implemented and experiments were conducted using Python `3.11.12`. The primary deep learning library utilized was PyTorch `2.6.0+cu124` (Paszke, 2019). Gaussian Process functionalities were implemented using GPyTorch `1.14` (Gardner et al., 2018). Other key libraries included NumPy `2.0.2`, SciPy `1.14.1`, and Scikit-learn `1.6.1`. The experiments were predominantly run on Google Colaboratory (Colab). The primary compute hardware utilized for experiments was a single `NVIDIA L4` GPU,equipped with approximately `23 GiB` of VRAM (23034 MiB). We acknowledge the use of Large Language Models (LLMs) to assist with generating boilerplate code, suggesting edits, and aiding debugging tasks during the development process- all strictly reviewed for correctness.

### S.G.2 BASELINE AND ABLATION MODEL DETAILS

**Convolutional Neural Network (CNN):** We implemented a CNN baseline by exactly replicating the "Systems Identification" branch architecture presented in (Qiu et al., 2023b), selected because it represents a well-validated model for this specific dataset. The architecture consists of two layers: a convolutional layer employing $9 \times 9$ kernels, followed by a fully connected layer. Consistent with (Qiu et al., 2023b), L2 regularization was applied to the convolutional filters to encourage smooth, center-surround-like receptive fields, while L1 regularization was applied to the weights of the fully connected layer to promote sparse feature usage for each output neuron. Hyperparameters were determined through grid search. The search space spanned: number of convolutional channels $\in \{8, 16, 24, 32\}$, L2 regularization coefficient $\in \{0, 0.1, 1.0, 10.0\}$, L1 regularization coefficient $\in \{0, 10^{-3}, 10^{-2}, 1/16, 0.1\}$, and learning rate $\in \{10^{-2}, 3 \times 10^{-3}, 10^{-3}, 3 \times 10^{-4}, 10^{-4}\}$. These ranges were based on the values used in the original work (Qiu et al., 2023b). Each model variant was trained for 12 epochs using the Adam optimizer ($\beta_1 = 0.99, \beta_2 = 0.999$) with a batch size of 32.

Training incorporated early stopping based on performance evaluated on a held-out validation/query dataset.

**Standard RBF Gaussian Process (GP):** The Radial Basis Function (RBF) GP baseline was implemented using the identical meta-adaptation script employed for our main model and ablation studies, omitting the feature extraction components. To ensure comparability, the same random seeds were utilized, acknowledging the sensitivity of non-parametric models to the specific conditioning data. GP hyperparameters (kernel lengthscale and variance) were optimized by maximizing the marginal likelihood. We initialized the lengthscale using the median heuristic and employed a broad prior variance (100) to avoid unduly constraining the lengthscale learning process. The model was trained for 300 epochs using the Adam optimizer with a learning rate of $4 \times 10^{-3}$ and betas $(\beta_1 = 0.99, \beta_2 = 0.999)$. Early stopping was triggered based on the correlation coefficient measured on the held-out validation/query set.

**Linear-Nonlinear (LN) Model:** This baseline models neural responses $\hat{\mathbf{y}}$ based on flattened input images $\mathbf{x}$ (vectorized size $36 \times 32 = 1152$) according to the function:

$$\hat{\mathbf{y}} = g(\mathbf{W}\mathbf{x} + \mathbf{b})$$

In this formulation, $\mathbf{W}$ denotes the weight matrix mapping the input vector to the 86 output neurons, $\mathbf{b}$ represents a bias vector, and $g$ is a pointwise nonlinearity. We employed the hyperbolic tangent $(tanh)$ function for $f$. Model parameters ($\mathbf{W}$ and $\mathbf{b}$) were learned by minimizing the Mean Squared Error (MSE) loss between the predicted responses $\hat{\mathbf{y}}$ and the ground truth responses $\mathbf{y}$, utilizing the Adam optimizer. L2 regularization was applied to the weight matrix $\mathbf{W}$. Hyperparameters were selected via grid search, optimizing the learning rate (6 values logarithmically spaced within $[5 \times 10^{-4}, 10^{-1}]$) and the L2 regularization coefficient (14 values logarithmically spaced from $10^{-6}$ to $10^1$ and 0). The optimal hyperparameter combination was chosen and early stopping triggered based on the highest correlation coefficient achieved on a held-out validation set. The model was trained for 400 epochs with a batch size of 20, using Adam with betas $(\beta_1 = 0.99, \beta_2 = 0.999)$.

**Ablation 1: Identity Deep Kernel** This ablation serves to isolate the contribution of the meta-learned deep feature extractor. It removes the feature extractor entirely, directly applying the task-adaptive linear heads ($h_i$) to the raw input pixels $\mathbf{x}$. The output features from the linear head are then fed into the task-specific RBF GP layer. The task-adaptive parameters (linear head weights $W_i$ and GP hyperparameters $\theta_{gp,i}$) were optimized for each task using the exact same adaptation procedure, hyperparameters, seeds, and initialization strategies described in Section S.F.3, ensuring consistency with the main 'Informed' model.

**Ablation 2: Random Deep Kernel** This ablation acts as a crucial control for the meta-learning process, testing the benefit of the theory-informed initialization against a random one, given the same fixed-extractor adaptation framework. It uses the same deep kernel architecture as the 'Informed' model (feature extractor $\phi$, task-adaptive linear heads $h_i$, RBF GP layer). However, the weights of the feature extractor $\phi$ were initialized randomly using the same scheme as the meta-learning initialization, but without loading the pre-trained weights. Crucially, mirroring the procedure for the 'Informed' model during task adaptation on the biological data, these randomly initialized feature extractor weights were kept frozen. Only the task-adaptive parameters (linear head weights $W_i$ and GP hyperparameters $\theta_{gp,i}$) were optimized using the exact same adaptation procedure, hyperparameters, seeds, and initialization strategies described in Section S.F.3. This ensures a direct comparison of the utility of the frozen meta-learned features versus frozen random features within our adaptation framework.

## S.G.3 RUNTIMES

We did not maintain precise runtime logs during experimentation; the following estimates are based on file timestamps, run on a single NVIDIA L4 GPU. The full meta-training process for our proposed model, including intermediate evaluations, required an estimated 3.5-4.5 hours. Generating the results presented for our model in Figure 3a (which involved evaluations across 10 training set sizes, for all 86 neurons) necessitated a total estimated runtime between 6 and 12 hours. The RBF GP baseline required total computational times comparable to our full informed model ( 6-12 hours). While the RBF GP lacks neural network computations during adaptation, its runtime may be influenced by factors such as kernel matrix conditioning. The CNN baseline required approximately 5-10 minutes for final training, not counting the hyperparameter tuning time. Runtimes for parametric

and nonparametric models used here are not directly comparable. The CNN predicts responses for all 86 neurons in parallel, while the adaptation phase for the GP models treats each neuron as a separate task with a 1D output. We briefly outline some **pathways toward scaling the framework to more neurons and larger datasets.** The meta-adaptation process is embarrassingly parallel, scaling linearly with the number of tasks (neurons). Our reported runtimes do not exploit this parallelism. Within each task adaptation, CNN forward/backward passes scale approximately linearly in the support set size $N$, whereas exact GP marginal log-likelihood (MLL) computations scale as $\mathcal{O}(N^3)$ time and $\mathcal{O}(N^2)$ memory because of the Cholesky factorization of the $N \times N$ kernel matrix. For applications with large support set sizes ($N$), standard GP approximations like variational inducing points are recommended (Titsias, 2009; Hensman et al., 2013; Ober et al., 2024).

## S.H    PROTOTYPE EXTRACTION AND INTERPRETABILITY ANALYSIS

This section provides further details on the method used to extract and analyze prototype images, offering insight into the task-specific representations learned by the linear heads of our deep kernel Gaussian Process model.

### S.H.1    COMPUTATION OF PROTOTYPE IMAGES

*For visual clarity, we opted to use the superscript $(i)$ here to denote task-indices, rather than the standard subscript $i$ throughout*

The "Prototype Image" interpretation method relies on analyzing how the task-specific linear heads $h^{(i)}$ (with parameters $W_i$), modify the geometry of the data representation learned by the shared feature extractor, $\phi$. We start by defining the pairwise Euclidean distances between input images $x_j$ and $x_k$ (vectorized) in the input space:

$$D_{jk}^{(x)} \;=\; \left\| x_j - x_k \right\|_2.$$

Images are then passed through the shared feature extractor $\phi(\cdot)$:

$$\phi_j \;=\; \phi(x_j).$$

The pairwise distances are computed in this shared feature space:

$$D_{jk}^{(\phi)} \;=\; \left\| \phi_j - \phi_k \right\|_2.$$

Subsequently, for each task $i$, the features are transformed by the task-specific linear head $h^{(i)}$:

$$h_j^{(i)} \;=\; W_i \phi_j.$$

The pairwise distances are computed again in this task-specific representation space:

$$D_{jk}^{(i)} \;=\; \left\| h_j^{(i)} - h_k^{(i)} \right\|_2.$$

As a practical implementation detail, before computing the difference matrix $\Delta_{jk}^{(i)}$, the distance matrices $D_{jk}^{(\phi)}$ and $D_{jk}^{(i)}$ are normalized (by dividing by their respective maximum absolute values) to ensure they are on a comparable scale.

The core idea is to quantify how the task-specific head $h^{(i)}$ changes the relative distances between pairs of images compared to the shared feature space. This change is captured by the difference matrix $\Delta_{jk}^{(i)}$:

$$\Delta_{jk}^{(i)} \;=\; D_{jk}^{(\phi)} \;-\; D_{jk}^{(i)}.$$

This matrix indicates pairs of images whose relative distance decreased ($\Delta_{jk}^{(i)} > 0$) or increased ($\Delta_{jk}^{(i)} < 0$) due to the task-specific transformation $h^{(i)}$.

To isolate the effect relative to the average change induced by the head for a given image $j$, we zero-center the difference matrix row-wise (i.e., for each fixed image $j$):

$$\widetilde{\Delta}_{jk}^{(i)} \;=\; \Delta_{jk}^{(i)} - \frac{1}{N} \sum_{k'=1}^{N} \Delta_{jk'}^{(i)},$$

where $N$ is the total number of images considered in the analysis. For our result we use $N = 750$.

Next, to relate these distance changes back to the input space, we compute a pixel-wise *overlap function*, $O_{jk}(u)$, between each pair of images $x_j$ and $x_k$. This function measures the similarity of pixel intensities $a_j(u)$ and $a_k(u)$ at each pixel location $u$, using a Gaussian weighting:

$$O_{jk}(u) \; = \; \exp\Bigl[-\frac{\bigl(a_j(u)-a_k(u)\bigr)^2}{2\sigma^2}\Bigr].$$

Here, $\sigma$ is a bandwidth hyperparameter controlling the sensitivity to pixel intensity differences, and we use $\sigma = 0.01$, but the method yielded similar results for other small values of $\sigma$ explored

For each image $j$ and each task $i$, we compute a *per-image contribution* map, $C_j^{(i)}(u)$. This map represents, for image $j$, the average pixel-wise overlap with all other images $k$, weighted by the normalized change in pairwise distance $\widetilde{\Delta}_{jk}^{(i)}$ induced by the task-specific head $h^{(i)}$. The normalization ensures that the contribution is scaled relative to the total magnitude of distance changes involving image $j$:

$$C_j^{(i)}(u) \; = \; \frac{\displaystyle\sum_{j\neq k} \widetilde{\Delta}_{jk}^{(i)}\, O_{jk}(u)}{\displaystyle\sum_{j\neq k} |\widetilde{\Delta}_{jk}^{(i)}| + \epsilon},$$

where $\epsilon$ is a small constant (e.g., $10^{-8}$) added for numerical stability.

Finally, the overall *prototype image* $P^{(i)}(u)$ for task $i$ is obtained by averaging these per-image contributions across all $N$ images:

$$P^{(i)}(u) \; = \; \frac{1}{N}\sum_{j=1}^{N} C_j^{(i)}(u).$$

This resulting image $P^{(i)}(u)$ can be interpreted as visualizing the average input pattern that leads to the largest relative changes in representation distances specific to task $i$.

### S.H.2 ALTERNATIVE NORMALIZATIONS FOR COMPUTING PROTOTYPES

Though the results presented in this work are derived using the difference between maximum-normalized distances heuristic, an approach more robust to extreme distance values would be to normalize $\Delta_{jk}$ by $D_{jk}^{(\phi)}$ rather than normalizing each $D$ by it's maximum. This would define an alternative $\Delta'$:

$$\Delta'_{jk}^{(i)} \; = \; \frac{D_{jk}^{(\phi)} - D_{jk}^{(i)}}{D_{jk}^{(\phi)} + \varepsilon},$$

Yet another method would be to first fit a scalar Procrustes factor $\alpha_i^*$ aligning the two distance matrices in the least-squares sense,

$$\alpha_i^* \; = \; \arg\min_\alpha \bigl\|D^{(\phi)} - \alpha D^{(i)}\bigr\|_F^2 \; = \; \frac{\sum_{j<k} D_{jk}^{(\phi)}\, D_{jk}^{(i)}}{\sum_{j<k}\bigl(D_{jk}^{(i)}\bigr)^2},$$

and then define

$$\Delta''_{jk}^{(i)} \; = \; D_{jk}^{(\phi)} - \alpha_i^* D_{jk}^{(i)}.$$

Our present results are robust to this normalization (Fig. S7), so we report both for reference but use the standard $\Delta_{jk}^{(i)}$ as defined in the section above.

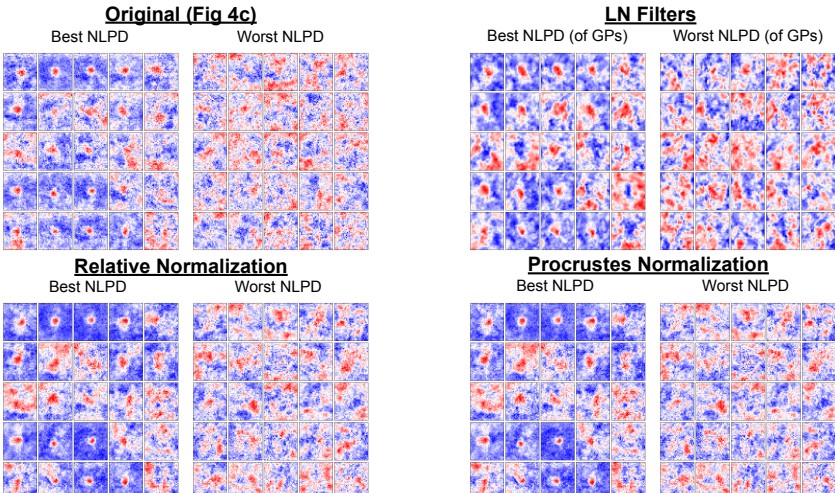

Figure S7: Prototype images obtained using Relative and Procrustes normalization, alongside the original max normalization used in the paper, and corresponding LN-model filters obtained by fitting on the biological data.

### S.H.3    ADDITIONAL VISUALIZATIONS

Here we extend the visualizations presented in Figures 4a and 4c.

**Extended 4a Synthetic Data Prototypes:** Figure S8 presents additional examples comparing the extracted prototype images $P^{(i)}(u)$ with the ground truth linear filters for the synthetic dataset experiments.

**Extended 4c Biological Neuron Prototypes:** Figure S9 displays prototype images $P^{(i)}(u)$ for a more extensive set of biological neurons beyond the best and worst 25 examples shown in Figure 4c.

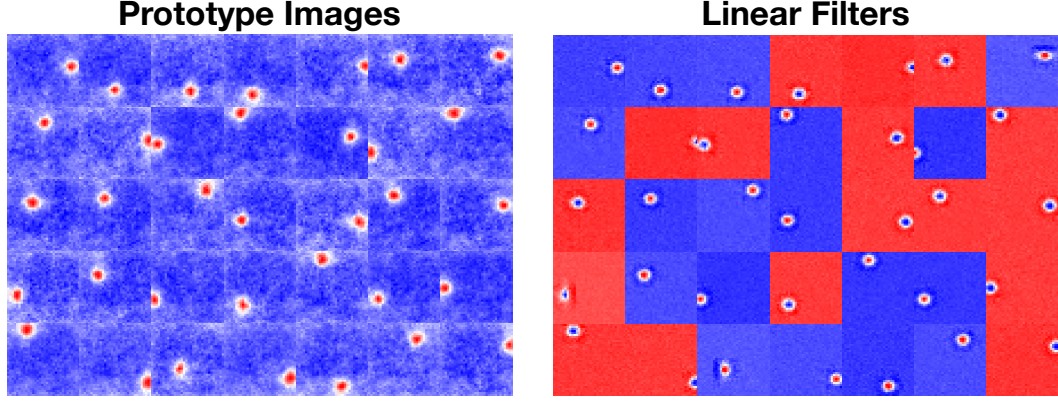

Figure S8: Additional examples comparing extracted prototypes $P^{(i)}(u)$ to ground truth linear filters for synthetic data.

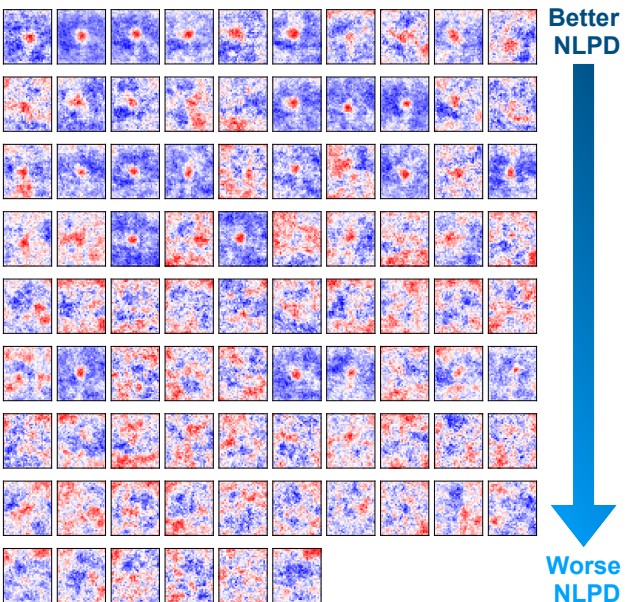

Better
NLPD

Worse
NLPD

Figure S9: Prototype images $P^{(i)}(u)$ for a broader selection of biological neurons, ordered by performance (NLPD) showing trends consistent with those of figure 4c

## S.I BAYESIAN MODEL COMPARISON AND OPTIMALITY VALIDATION

### S.I.1 ABLATION STUDY ON BAYESIAN MODEL COMPARISON

To confirm that our method is specifically sensitive to the theory's structure, we applied our ablation conditions on our Theory Informed model for the Bayesian Model comparison analysis. Here, we report the Pearson correlation coefficients obtained by implementing the exact same "identification of optimality level" protocol using the "Random", "Identity", and "Task Ablation" conditions.

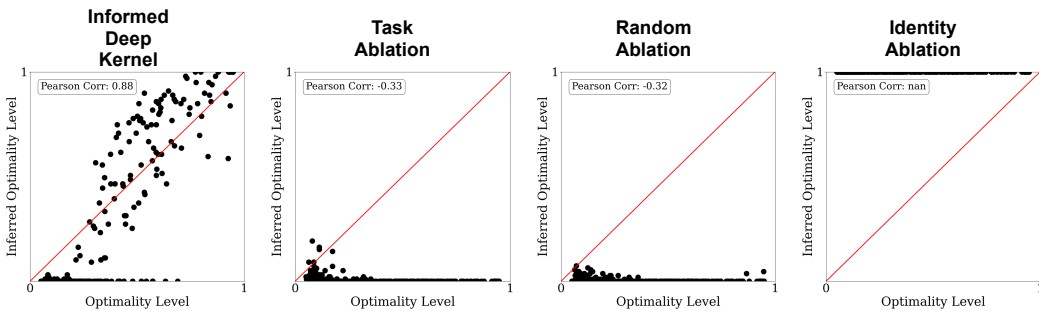

Figure S10: Ablation Study on Bayesian Model Comparison shows Theory Informed Features indispensable

When using the unstructured prior, the inferred optimality level is either anti-correlated (Pearson corr. = -0.32, Pearson corr. =-0.33 ) with ground truth, or simply not sensitive at all (Figure S10). This confirms that the model comparison capability is driven specifically by the theory/data match.

### S.I.2 MIXTURE KERNEL FORMULATION

To smoothly interpolate between a full explanation by the normative theory and a null hypothesis, we define a mixture kernel $K_\beta$ as a convex combination of the theory-informed kernel $K_{\text{TIK}}$ and a

standard Radial Basis Function (RBF) kernel $K_{\text{RBF}}$:

$$K_\beta(x, x') = \beta K(x', x)_{\text{TIK}} + (1 - \beta)K(x', x)_{\text{RBF}}$$

Here, $\beta \in [0, 1]$ is a mixing coefficient that represents the degree of belief allocated to the theory-informed model. $K_{\text{TIK}}$ encodes the prior assumptions derived from the specific normative theory being tested (e.g., efficient coding leading to center-surround receptive fields). The RBF kernel, defined as $K_{\text{RBF}}(x, x') = \theta_0^2 \exp\left(-\frac{\|x-x'\|^2}{2\ell^2}\right)$ with outputscale $\theta_0$ and lengthscale $\ell$, serves as the null model.

Our rationale for using the RBF kernel as the null model is that it represents a flexible, general-purpose prior over smooth functions (Williams & Rasmussen, 2006), and that the Deep Kernel model itself is an RBF GP operating on top of a learned feature space. The RBF makes minimal structural assumptions about the data-generating process beyond smoothness, controlled by its hyperparameters $(\theta_0, \ell)$. Therefore, it provides a reasonable baseline against which the specific structural assumptions encoded in $K_{\text{TIK}}$ can be compared. A higher inferred $\beta^*$ indicates that the specific structure imposed by the normative theory provides a better explanation for the data than the generic smoothness assumption of the RBF kernel.

## S.I.3    INFERENCE OF OPTIMALITY LEVEL ($\beta^*$)

We infer the optimal mixing coefficient $\beta^*$ by maximizing the marginal likelihood of the Gaussian Process model using the mixture kernel $K_\beta$. The marginal likelihood $P(Y|X, \beta)$ represents the probability of observing the neural responses $Y$ given the stimuli $X$ and the specific mixed prior defined by $\beta$. The optimal $\beta^*$ is found by:

$$\beta^* = \arg \max_{\beta \in [0,1]} P(Y|X, \beta)$$

This maximization is performed by evaluating the marginal likelihood across a discretized grid of $\beta$ values. Specifically, we evaluated $P(Y|X, \beta)$ for 100 evenly spaced values of $\beta$ ranging from 0 to 1.

Crucially, before evaluating the marginal likelihood across the $\beta$ grid, other hyperparameters of the model are optimized and then frozen. This includes:

1. Task Adaptive parameters of the informed kernel $K_{\text{TIK}}$, (i.e. linear head weights and GP layer hyperparameters).
2. Hyperparameters of the RBF kernel $K_{\text{RBF}}$ (i.e. outputscale $\theta_0$, lengthscale $\ell$, and noise).

These parameters are learned by maximizing the marginal likelihood of their respective base models Once optimized, these parameters are held fixed during the grid search for $\beta^*$. This ensures that the comparison focuses specifically on the contribution of the informed structure versus the null structure, as interpolated by $\beta$ (Młynarski et al., 2021).

## S.I.4    SYNTHETIC VALIDATION DETAILS

To validate our Bayesian model comparison approach for inferring optimality, we first applied it to synthetic data where the "ground truth" level of optimality could be controlled and measured independently.

**Generating Controlled Optimality Receptive Fields (RFs)**    We heuristically generated synthetic receptive fields (RFs) with varying degrees of optimality using a controlled noise-injection process. The procedure involved the following steps:

1. **Initial Optimal RFs:** We started with a set of 20 archetypal synthetic "optimal" RFs.
2. **Generating "Anti-Optimal" Noise:** To perturb these RFs away from optimality in a structured manner, we generated a basis for noise that is explicitly orthogonal to any possible center-surround receptive fields like ours. This was achieved by:
   - Taking a larger reference set of 4000 theoretically "optimal" center-surround RFs (distinct from the initial 20 RFs and any meta-training data). Let this set be $J_{RF} \in \mathbb{R}^{4000 \times H \times W}$. We also considered the transposed versions $J_{RF}^T$.

- Flattening these RFs into vectors $J_{\text{flat}} \in \mathbb{R}^{8000 \times (H \cdot W)}$.
- Performing Singular Value Decomposition (SVD) on $J_{\text{flat}}$: $J_{\text{flat}} = USV^T$.
- Determining the effective rank $r$ of the optimal RF subspace by thresholding the singular values $S$. We used a threshold ratio of $0.1$ relative to the maximum singular value.
- Identifying the subspace orthogonal to the dominant $r$ principal components (columns of $V$). The projection matrix onto the optimal subspace is $P = V_r V_r^T$, where $V_r$ contains the first $r$ columns of $V$. The projection onto the orthogonal subspace is $P_\perp = I - P$.
- Generating noise images by projecting natural images ($N_{\text{img}}$) onto this orthogonal subspace: $\text{Noise}_{\text{img}} = P_\perp N_{\text{img}}$. These images represent patterns or structures that are explicitly *anti* optimal. Figure S11 shows examples of such generated noise components.

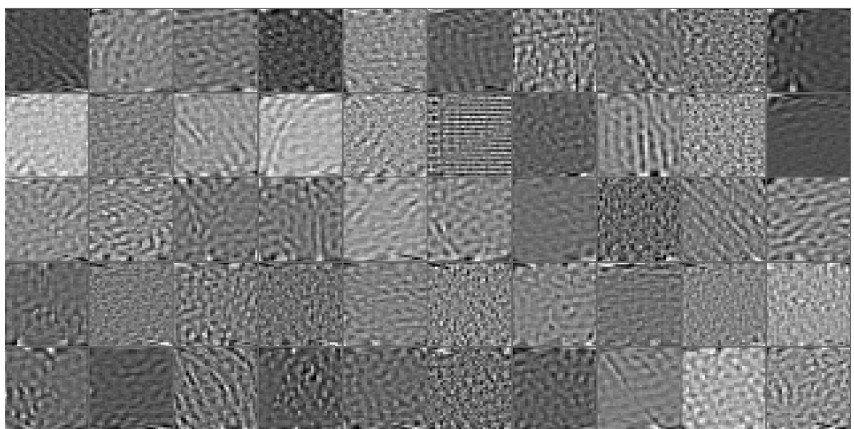

Figure S11: Example "Noise" images used for generating suboptimal RFs. These images were generated by finding the component of natural images orthogonal to the subspace spanned by a large set of optimal center-surround receptive fields.

3. **Perturbation Process:** We simulated a random walk for each initial optimal RF over $T$ timesteps (Here, T=600). At each timestep $t$:
    - We sample a random "anti-optimal" noise image.
    - This noise image was scaled by a random Gaussian scalar $z \sim \mathcal{N}(0, \text{scale})$, where scale = 0.01 for our purposes.
    - The scaled noise was added to the RF from timestep $t-1$: $RF_t = RF_{t-1} + z \cdot \text{Noise}_{\text{img}}$.
    - The resulting $RF_t$ was normalized by subtracting its mean and dividing by its L2 norm: $RF_t = (RF_t - \overline{RF}_t)/\|RF_t\|_2$. This process gradually pushes the RF away from its initial optimal structure .

4. **Subsampling Trajectories:** The full perturbation process generates long trajectories (T=600 steps) for each initial RF. To create a dataset with distinct, relatively stable levels of suboptimality, we subsampled these trajectories. For each RF's trajectory of ground truth $R^2$ values (Sec. S.I.4), we selected a smaller number of points (20). The selection aimed to capture points along a smooth, approximately linear decay in $R^2$ by minimizing the residual from a global linear fit within predefined index windows. This ensures the final synthetic dataset contains RFs spanning a range of controlled optimality levels, while avoiding pathological points, for instance when the DoG model fails to fit and produces a nonsense $R^2$ value.

**Ground Truth Optimality Quantification:** $R^2$ **of DoG**    To obtain an independent, "ground truth" measure of optimality for the generated synthetic RFs, we used the standard approach of fitting a Difference of Gaussians (DoG) model to each RF and calculating the coefficient of determination ($R^2$). This is motivated by the theoretical understanding that optimal RFs under certain efficient coding assumptions often resemble DoG functions (Atick & Redlich, 1992).

1. **DoG Model:** The DoG model is defined as the difference of two concentric Gaussian functions with different amplitudes and standard deviations:

$$\text{DoG}(x, y|\theta) = A_c \exp\left(-\frac{(x - x_0)^2 + (y - y_0)^2}{2\sigma_c^2}\right) - A_s \exp\left(-\frac{(x - x_0)^2 + (y - y_0)^2}{2\sigma_s^2}\right)$$

   The parameter vector is $\theta = \{A_c, A_s, x_0, y_0, \sigma_c, \sigma_s\}$, representing the amplitudes of the center and surround Gaussians, their common center $(x_0, y_0)$, and their standard deviations $(\sigma_c, \sigma_s)$.

2. **Fitting Procedure:** For each synthetic RF (dimensions $D1 \times D2$), we fitted the DoG model by minimizing the mean squared error between the flattened RF pixel values ($RF_{flat}$) and the DoG model output evaluated on a corresponding grid ($DoG_{flat}(\theta)$):

$$\hat{\theta} = \arg\min_{\theta} \frac{1}{D1 \cdot D2} \|\text{RF}_{\text{flat}} - \text{DoG}_{\text{flat}}(\theta)\|^2$$

3. **Initialization:** Parameter initialization significantly impacts DoG fitting. We used the following strategy:

   - The scale of the center, $A_c$ was initialized to the maximum value of the RF.
   - The scale of the Surround, $A_s$ was initialized to $0.5 \times A_c$.
   - $\log \sigma_c$ and $\log \sigma_s$ were initialized to fixed values corresponding to $\sigma_c \approx 3.0$ and $\sigma_s \approx 6.0$.
   - The center location $(x_0, y_0)$ was initialized using a localized center-of-mass heuristic. This involved calculating 1D summed projections of the RF along rows and columns. A 1D convolution with a sliding window was applied to these projections to find the region of maximal energy. The center of mass was then computed within this window. To improve robustness, this process was repeated for multiple window sizes . The DoG fit was performed for each resulting $(x_0, y_0)$ initialization, and the fit yielding the highest final $R^2$ was selected.

4. $R^2$ **Calculation:** After fitting, the goodness-of-fit was quantified using the coefficient of determination ($R^2$):

$$R^2 = 1 - \frac{\sum_{i=1}^{D1 \cdot D2}(\text{RF}_{\text{flat},i} - \text{DoG}_{\text{flat},i}(\hat{\theta}))^2}{\sum_{i=1}^{D1 \cdot D2}(\text{RF}_{\text{flat},i} - \overline{\text{RF}}_{\text{flat}})^2}$$

   where $\overline{\text{RF}}_{\text{flat}}$ is the mean value of the RF pixels. This $R^2$ value serves as the "ground truth" optimality measure for each synthetic RF, against which our inferred $\beta^*$ values were compared (in Figure 5a of the main text).

## S.J  GLOBAL HYPERPARAMETER TABLE

Table S6: Summary of Key Hyperparameters

| Parameter Category | Hyperparameter | Value / Setting |
|---|---|---|
| **Meta-Training** | Optimizer | Adam |
| | Total Epochs ($E$) | 7 |
| | Meta Batch Size ($B$, num tasks) | 10 |
| | Total Meta-Train Tasks | 490 (49 batches $\times$ 10 tasks) |
| | Task Data Points (Support + Query) | 1452 |
| | Support/Query Split | 5% Support / 95% Query (resampled each epoch) |
| | Inner Adaptation Steps ($n_{\text{inner steps}}$) | 17 |
| | Inner LR - Linear Head ($\alpha_{\text{linear}}$) | $4 \times 10^{-4}$ (0.0004) |
| | Inner LR - GP Params ($\alpha_{\text{GP}}$) | $2 \times 10^{-3}$ (0.002) |
| | Outer Meta-Update Steps ($n_{\text{outer steps}}$) | 3 |
| | Outer LR ($\alpha_{\text{outer}}$) | $4 \times 10^{-4}$ (0.0004) |
| | Adam Betas ($\beta_1, \beta_2$) | (0.5,0.5) |
| | Gradient Clipping (Feature Extractor Norm) | 1.0 |
| | Initial LR Scaling (First Epoch) | 1/10th |
| | GP Noise $\sigma_\eta$ (Synthetic Tasks) | $10^{-4}$ |
| | Data Type | `torch.double` |
| **Task Adaptation** | Adapted Components | Linear Heads ($h_i$), GP Hyperparameters ($\theta_{gp,i}$) |
| | Frozen Components | Feature Extractor ($\phi$, both Informed and Random) |
| | Optimizer | Adam |
| | Epochs | 300 |
| | Base Learning Rate (for GP Params) | $4 \times 10^{-3}$ |
| | Linear Head LR Scale | $10^{-2}$ (Effective LR: $4 \times 10^{-5}$) |
| | Adam Betas ($\beta_1, \beta_2$) | (0.99, 0.999) |
| | L1 Sparsity (Linear Heads) | $10^{-2}$ |
| | GP Lengthscale Prior | Gaussian(mean=Median Init, var=0.01) |
| | GP Lengthscale Init | Median Heuristic |
| | GP Noise $\sigma_\eta$ Init (Biological) | Standard GPyTorch Initialization |
| | GP Noise $\sigma_\eta$ Init (Synthetic Test) | $10^{-4}$ |
| **CNN Baseline** | Architecture | Qiu et al. (Qiu et al., 2023b) "SI Branch" |
| | Optimizer | Adam |
| | Adam Betas ($\beta_1, \beta_2$) | (0.99, 0.999) |
| | Epochs | 12 |
| | Batch Size | 32 |
| | Hyperparameter Selection | Grid Search (Val. Correlation) |
| | Grid: Conv Channels | {8, 16, 24, 32} |
| | Grid: Conv L2 Reg | {0, 0.1, 1.0, 10.0} |
| | Grid: FC L1 Reg | {0, $10^{-3}$, $10^{-2}$, 1/16, 0.1} |
| | Grid: Learning Rate | {$10^{-2}$, $3 \times 10^{-3}$, $10^{-3}$, $3 \times 10^{-4}$, $10^{-4}$} |
| **RBF GP Baseline** | Adapted Components | GP Hyperparameters ($\theta_{gp,i}$) |
| | Optimizer | Adam |
| | Epochs | 300 (with Early Stopping on Val. Correlation) |
| | Learning Rate | $4 \times 10^{-3}$ |
| | Adam Betas ($\beta_1, \beta_2$) | (0.99, 0.999) |
| | Lengthscale Init | Median Heuristic |
| | Lengthscale Prior Variance | 100 (Broad) |
| **LN Baseline** | Non-linearity ($g$) | tanh |
| | Optimizer | Adam |
| | Adam Betas ($\beta_1, \beta_2$) | (0.99, 0.999) |
| | Epochs | 400 |
| | Batch Size | 20 |
| | Hyperparameter Selection | Grid Search (Val. Correlation) |
| | Grid: Learning Rate | 6 values log-spaced [$5 \times 10^{-4}$, $10^{-1}$] |
| | Grid: L2 Reg (Weights $\mathbf{W}$) | 14 values log-spaced [$10^{-6}$, $10^1$] + [0] |

