# OpenReview forum: "Meta-Learning Theory-Informed Inductive Biases using Deep Kernel Gaussian Processes"
_ICLR.cc/2026/Conference — ICLR 2026 Poster_

### Official Review · Reviewer_JnnY · 2025-10-27

**Soundness:** 3
**Presentation:** 3
**Contribution:** 3
**Rating:** 6
**Confidence:** 1

**Summary:**

I'm very sorry, but even after three readings I did not understand this paper; it is simply _way_ out of my area of expertise. I'm going to give it a neutral review: goods all around, with the lowest possible confidence. But this review should be totally discarded.

**Strengths:**

n/a; see above.

**Weaknesses:**

n/a; see above.

**Questions:**

If the paper were totally rewritten in such a way that I could understand it, the would help. But it's hardly an appropriate task for the authors. However, if they want to have impact on neuroscientists (at least neuroscientists such as myself), they're going to have to publish a massively dumbed-down version.

---

> ### Author Response · Authors · 2025-11-20
>
> Dear Reviewer JnnY,
>
> Thank you for your honesty and for making the effort to read the paper multiple times despite it being outside your main area of expertise. We appreciate your comment about accessibility to a broader neuroscience audience.
>
> A much more accessible exposition aimed specifically at a general neuroscience audience would likely require a different, longer format (e.g. a review or journal article), and we do hope to pursue that in follow-up work.
>
> Thanks again for taking the time,
>
> The Authors

---

> > ### Comment · Reviewer_JnnY · 2025-11-20
> > **reply from reviwer JnnY**
> >
> > You're welcome. ;) By the way, I did contact the area chair, and my review is not being counted. Probably pretty obvious, but I thought I should make it crystal clear.

---

### Official Review · Reviewer_SKqL · 2025-10-30

**Soundness:** 3
**Presentation:** 4
**Contribution:** 3
**Rating:** 8
**Confidence:** 4

**Summary:**

This research presents a new Bayesian machine learning framework to bridge the gap between theoretical neuroscience and data analysis. The method automatically translates complex “normative” theories (like efficient coding) into practical, probabilistic models that can be directly used to analyze biological data. By using a “theory-informed kernel” within a deep Gaussian process, the framework improves prediction accuracy, provides reliable uncertainty estimates, and allows for rigorous testing of the underlying theory – all without requiring manual tuning or specific parameterizations of the theory itself.

The work delivers a scalable and automated way to integrate theoretical knowledge into data-driven neuroscience, demonstrating improved performance on retinal ganglion cell data. Beyond neuroscience, it also advances Bayesian machine learning by providing a novel application of deep kernels for domain-informed model design, addressing common challenges like “feature collapse” to ensure well-calibrated uncertainty. This framework ultimately allows researchers to not only use theories to understand data, but also test and refine those theories with data in a statistically rigorous manner.

**Strengths:**

This work demonstrates considerable merit through a novel and conceptually strong approach to integrating theoretical frameworks with data-driven analysis. The authors’ strategy of embedding inductive bias via the Theory-Informed Kernel, learned from simulations generated by normative theory, is particularly intriguing and offers a compelling pathway toward more biologically plausible modeling.

The methodological foundation built upon Bayesian machine learning is a significant strength, with the provision of calibrated uncertainty estimates representing a valuable addition to the standard machine learning toolkit. Furthermore, the proposed Bayesian model comparison framework appears to be a promising and well-motivated concept, facilitating rigorous hypothesis testing and quantitative evaluation of theoretical predictions.

Finally, the inclusion of detailed technical considerations regarding Deep Kernel Learning (DKL) training, as outlined in Supplementary Figure S.F.4, constitutes a valuable contribution to the field, offering practical insights and potentially mitigating common pitfalls such as feature collapse as previously reported in the literature. This level of technical detail enhances the reproducibility and broader applicability of the presented methodology.

**Weaknesses:**

The paper appears to be written mainly by neuroscience domain experts,
making it difficult for readers primarily interested in the ML methodology
to follow due to the use of domain-specific terminology. For instance, the
manuscript frequently employs neuroscience-specific terms without introduction
-- for example, the use of “bottleneck” in expressions such as “minimizing
bottleneck activity” (line 215) and “single optimized bottleneck neuron” (line
237) -- which may not be immediately clear to readers from the ML community.
Section 4.1 and the corresponding supplementary material are challenging to
interpret for non-experts; translating the neuroscience terminology into
machine learning concepts requires some effort before the underlying regression
setup becomes apparent. The interchangeable use of the terms “task” and “neuron”
is confusing and should be avoided.

**Questions:**

Content:

* The authors should elaborate on why it is necessary to have "task-specific"
  linear layers feeding into task-specific GP models. The text introduces the
  fact but doesn't actually seem to motivate it.
* Line 266: The authors could elaborate on how the "base RBF GP" works when the
  feature extractor is not used, i.e. how are the input images processed and
  fed to the linear head and GP in this case?
* Fig. 3b, Sec. 4.4.: the authors should elaborate on the principal components:
  how many PCs are needed to faithfully explain the data / why is it adequate
  to show the GP prediction as function of just PC1?
* Fig. 3b, Sec. 4.4.: from the plots it seems that the DKL with RBF base kernel
  has similar behavior to the "bare" RBF GP baseline - in both cases the RBF's
  stiffness / smoothness / low curvature seems to dominate and the model
  doesn't actually seem to model the data particularly well (esp. in the 1400
  points case) -- the "informed" kernel's effect seems minor.
* Fig. 5a: The authors should explain the tight cluster of points parallel to
  the x-axis, they don't seem to be part of the linear trend point cloud.
* The manuscript introduces "support" and "query" data (line 140) and equate
  those to more familiar "train" and "validation" data, which would make the
  former two terms unnecessary. Please explain why this new terminology is
  needed.
* In several places the authors mention "well-calibrated (epistemic)
  uncertainty". Around lines 1446 and 1589 it is mentioned that the
  likelihood's noise level is set close to zero. Typically in regression
  settings, GPs are calibrated when the total uncertainty, in the simplest case
  of a scalar homoscedastic $\sigma_n$, $\Sigma + \sigma_n^2\,I$, is
  considered. If the likelihood noise is (assumed to be) zero, then the authors
  seem to suggest that the variance in the data is explained by model
  uncertainty alone. In this case, calibration plots or calibration error
  metrics (see e.g. http://arxiv.org/abs/2109.10254) should be reported to back
  the claim that uncertainties are indeed "well-calibrated".

- line 265: “the improved performance over the bare RBF GP indicates that the learned embedding is meaningful” such indirect assessments suggest to be glossing over details as the performance of the whole system is taken as an indicator for the performance of a subsystem. Please be more precise.

- line 271: “this subverts the conventional idea about kernel methods, such as GPs, being specialized for extremely low data regimes.” This reads like an overly general statement without references. I relate this statement to the approach documented in the paper, which cannot give rise for a general statement about GPs.

- line 288: “with both models fit on the full dataset” I hope that the models have only been fitted on the training set and a hold-out set or cross-validation was performed. Please reconsider this formulation.

- line 296: “this suggests … our representation usefully structures the input space” no evidence is given for this assessment, please consider removing it.

- line 299: the introduction of the main metric NLPD is very brief. The text does hardly detail where the probability distribution in question is obtained from. Please add details on how NLPD is computed in your use case. This provides a better angle for people in the ML community unfamiliar with this metric to judge your results.

- line 323: 'As N grows, the GP layers ... letting the data "speak for themselves".' Please remove this heuristic description of an effect for which the paper does not present evidence.

- page 7, fig 4.a. Please add a color scale, so that the different tones of red/blue can be interpreted

Notation:

* Table S1: please use scientific notation $A \times 10^{-B}$ vs. ``Ae-B``
* The manuscript shows inconsistent notation (or typos) in several places:
  * task $T$ (line 216) vs. $\mathcal T$ (line 161)
  * likelihood noise $\sigma_n$ (lines 1547, 1575) vs. $\sigma_\eta$ (line 127 and others)
  * $\theta_\text{gp}$ (line 124) vs. $\theta_{gp}$ (1375)
  * $exp$ (line 1375) vs. $\exp$
  * text in subscripts such as $\mathcal D_{train}$ (line 126) vs. $x_\text{train}$ (line 300)
  * $P(...)$ (line 404) vs. P$(...)$ (line 416)
* There is code-specific notation lacking corresponding typesetting, e.g.
  ZeroMean() instead of $\texttt{ZeroMean()}$ or better just "zero mean" in
  this case maybe (line 1375), Scikit-Learn’s NearestNeighbors instead of
  $\texttt{NearestNeighbors}$, same with n_clusters=None (Sec. S.D.3)
* line 404 and 416: although the formula in the letter line for $\beta^*$ is targetting the grid search, please check if the use of $\beta$ and $K_\beta$ is correct. If so, please comment why $K_\beta$ is used in the latter formula on line 416.

---

> ### Author Response · Authors · 2025-11-20
> **Reply to reviewer SKqL Part (1/2)**
>
> Dear Reviewer SKqL,
>
> Thank you for your highly detailed review that has helped us improve our manuscript.
> We particularly appreciated your very specific line-by-line suggestions that helped us tighten the language, improve readability, and correct typos where necessary.
>
>
> ## Re: the Notation corrections
> We’ve implemented all of your suggestions into the manuscript. The $K_\beta$ was indeed a typo, and we fixed that.
>
> Thank you for spotting these.
> ## Re: Weaknesses
> **Actions:**
>
> **i)** We have clarified in the text, for ML readers, that by bottleneck neuron we refer to the activity in the bottleneck layer of the autoencoder architecture (line 248, brown)
>
> **ii)** We have also added some explanation to Sec 4.1 for our ML readers (Line 239, brown)
>
> Regarding the interchangeable use of “task” and “neuron”, this we agree is not perfect phrasing, but we feel it's best to keep it, as it's the least bad option we could think of. If we stick to “tasks” only, we feel this would alienate our neuroscience readers. If we stick to “neurons” only, it neglects the fact that the neurons are meta-test tasks, and also is ambiguous about whether we refer to artificial neurons in a neural network or biological measurements.
>
> # Re: Content corrections/Questions
>
> ## Re: Both points about Fig. 3b
>
> **i)** One PC does not faithfully explain the data; substantially more PCs would be needed for that. PC1 in Fig. 3b is used only for qualitative visualization: a 1D slice that highlights the contrast between the RBF GP and the TIK. Along this slice, the TIK maintains non-collapsed epistemic uncertainty even in regions densely covered by data, whereas the RBF GP becomes overconfident. This suggests that we have avoided feature collapse, which, if severe, would manifest as uniformly low epistemic uncertainty that is insensitive to variation in the data (e.g. Figure 1b of Ober et al 2021; https://proceedings.mlr.press/v161/ober21a/ober21a.pdf)
>
> **ii)** In Fig. 3b, both the DKL model and the “bare” RBF GP produce smooth functions along PC1, which is expected since both ultimately use an RBF kernel at the GP layer. Near the main cluster of data around PC1 ≈ 0, the DKL prediction does bend more clearly downward toward the cluster of red data points, but we would not expect dramatic visual differences in a single 1D projection of a high-dimensional function: substantial variation may occur along other directions that are not visible in this slice. In addition, these biological measurements are noisy and affected by unobserved covariates, so the overall prediction ceiling is low; even modest apparent improvements can reflect a substantial fraction of the explainable variance. For these reasons we do not view Fig. 3b as the primary evidence for the benefit of the informed kernel. The substantive differences are better captured by our quantitative full-dimensional metrics in Fig. 3a,c, where the DKL model consistently improves both accuracy and NLPD

---

> > ### Author Response · Authors · 2025-11-20
> > **Reply to reviewer SKqL Part (2/2)**
> >
> > ## Re: Support and Query data
> > We agree that this can be confusing for readers not already familiar with meta-learning jargon. In the meta-learning literature, it is standard to distinguish between (i) tasks, which are split into meta-train / meta-val / meta-test sets, and (ii) within-task datasets, which are split into support and query sets. In our setting, we meta-train on synthetic tasks and meta-test on biological tasks, so we adopted this standard terminology to make clear that “support vs. query” refers to splits within a task, whereas “train/val/test” usually refers to splits between tasks or datasets. To aid intuitive understanding, we also deliberately mix the wording in a few places (e.g., in Fig. 2 and Sec. 4.1) to help readers map the meta-learning terms to the more familiar train/validation language.
> >
> > ## Re: “Well Calibrated Epistemic Uncertainty”
> > We agree that calibration in regression should be evaluated with respect to the total predictive uncertainty. The near-zero likelihood noise levels mentioned around lines 1446 and 1589 (now lines 1558 and 1700) apply only to the meta-training (which is done on genuinely noise-free synthetic tasks) and to Fig. 3b, where we intentionally visualize epistemic uncertainty only. In our main biological experiments and in Fig. 3c, the likelihood noise is treated as a standard GP hyperparameter and is learned from the data via marginal-likelihood optimization, so the predictive variance used in **the NLPD includes both epistemic and aleatoric components**.
> >
> > **Action:**
> >
> > **i)** we clarify in the text that we use the total uncertainty for our NLPD (Line 320, Brown)
> >
> > **ii)** to avoid overloading the term “calibrated,” we have *re*calibrated some of the phrasing to more precisely emphasize improved uncertainty quantification rather than formal calibration metrics (lines 24,102, Brown)
> >
> > ## Re: Tight cluster of points in Figure 5a
> > We have previously inspected the points the reviewer refers to and found that many of them are generated from the same few base receptive fields (RFs) whose centers lie very close to the edge of the image. Therefore we speculate that this may be an **edge artifact**.
> >
> > There are two ways this can transpire:
> > First, the TIK itself may be better suited at modelling full receptive fields than oddly shaped ones at the boundary, pushing $\beta^\*$ to $0$. Second, the smaller size of what remains in the RF may make it harder for the DoG model to accurately quantify the “True” optimality level due to optimization failures.
> >
> > The biological data, however, typically do not contain such “edge” RF’s because they would typically not make it past the signal-to-noise ratio screening (line 1258), so that may be one reason we don’t see the same artifacts in Fig 5b.
> >
> > ## Re: All Other points in Content
> > We generally agree with your suggestions and have implemented changes in the text to reflect them.
> >
> > **Actions:**
> > Re: elaborating on Task Specific linear layers, base RBF GP inputs, explaining NLPD, adding color scale,  tightening phrasing about “learned embedding meaningful”, “subverts conventional idea”, “both models fit on full dataset”,“usefully structures input space” ,”letting data speak for themselves”
> >
> > **We have made all of these suggested changes** in the manuscript (Lines 161,277,316,325,311,285,302,305,278,366 brown)
> >
> > ## Thank you again for your helpful and detailed review.

---

> > > ### Comment · Reviewer_SKqL · 2025-11-27
> > > **Thanks for the great rebuttal**
> > >
> > > Thanks so much for addressing concerns and recommendations. I am happy to keep my score at where it is.

---

### Official Review · Reviewer_Xmij · 2025-11-01

**Soundness:** 3
**Presentation:** 4
**Contribution:** 3
**Rating:** 8
**Confidence:** 4

**Summary:**

In this work the authors present a novel framework to capture normative theoreies as deep kernels for a Gaussian Process by framing the extraction process as meta-learning framework. Based on simulation results generated from a theoretical model, the inductive-bias of the theory is sought to be captured by learning a feature extractor that works across distinct instances of tasks, thereby serving as basis of GP kernel that in turn can be used to model and assess real experimental data. The Bayesian model nature of this approach (via the GP) allows for assessment typically difficult in theoretically-motivated models such as the degree of epistemic uncertainty and fidelity of the theory in capturing the experimental data.

**Strengths:**

Clarity: The work is clearly written with easy to follow structure. All necessary background materials were described suffiicently allowing for readers who may not be deeply familiar with the topic to follow and appreciate the contribution.

Quality: All derivations and mathematical motivations are clearly presented and accurate.


Originality & Significance
The paper addresses a critical challenge in computational neuroscience — quantitative and comparable assessment of alternative normative theories. As the authors highlight in introduction, the field has been in great need of a method to combine the inductive bias provided by normative theories while providing data-rooted assessment of the merits (or lack thereof) of theory-based inductive bias in improving model predictions.
The theory-informed Gaussian process introduced by the authors provide a straightforward approach  to possibly frame theoretical predictions into a meta learning task, capturing the effective inductive-bias into the meta-learned feature extractor. Unlike traditional theoretically-motivated approaches, this method provides for a concrete approach to turn theoretical-model with plausible simulation into ultimately a Gaussian process model that can be fit and evaluated on experimental data. The work is quite original with a theoretically well-motivated application of deep GP kernels in capturing the inductive bias through the formulation of meta-learning framework.

**Weaknesses:**

Although the bridge from the inductive-bias into meta-learning framework is elegant, the presented case of efficient coding appears quite limited, and leaves applications to a broader theoretical inductive-bias deserved to fully assess the generalizability of this approach. In particular, for the efficient coding example, it appears that meta learning tasks were all derived from a specific instance of auto-encoder model trained on natural images in accordance to efficient-coding objective. Given this, it appears to be an overstatment to me to say that the meta-learning framework can be said to have captured the inductive-bias of the full notion/theory of efficient coding. Rather, it appears that the Gaussian process feature extractor only effectively captured the very specific instance of an autoencoder, and thus not necessary can be deemed to be representative of the underlying theoretical framework.
It is also not clear

**Questions:**

Questions:
* Can this framework be applied to other categories of normative theories such as neural sampling code/theory?
* Would it be viable to meta-learn the feature extractor on multiple instances of theoretical models (e.g. such as models trained and optimized on a distinct set of natural images, but all still trained using EC objective)?

---

> ### Author Response · Authors · 2025-11-20
> **Reply to reviewer Xmij**
>
> Dear Reviewer Xmij,
>
> Thank you for taking the time to review our work. We are encouraged by your positive assessment, and would like to address some of the questions and weaknesses below.
>
> # Weaknesses
> **We agree** that, in our experiments, the meta-learned kernel captures the structure of a particular instantiation of efficient coding (a specific convolutional autoencoder trained on natural images), not the “full” notion of efficient coding in all its possible forms.
>
> **We did not intend to claim that the entirety of efficient coding theory is captured by this single inductive bias, and we completely agree that this would be inaccurate**. Efficient coding is, anyway, not a single monolithic model but a family of related objectives and architectures that can make different predictions in different regimes.
>
> **We actually view this specificity as a strength**. Our framework should, in principle, enable comparisons between different instantiations of the same normative idea, for example, an LN-based efficient coding model vs. an autoencoder-based one, within a common GP framework- even if they live in very different parameterizations.
>
> **We have adjusted wording** in the manuscript to avoid giving the impression of claiming to capture efficient coding in totality (Lines 19,77,84,87,280,496; edits in Violet)
>
>
> Thank you for pointing this out.
>
> # Questions
>
> ## Re: Application to neural sampling code
> **Yes.** Our framework only requires that a normative theory induces a (possibly stochastic) mapping from stimuli to responses that can be instantiated as a black-box predictor or simulator. Neural sampling code (NSC) theories do exactly this: they posit that neural activity implements sampling from a generative model over latent variables underlying sensory inputs, so for each stimulus $x$ they define a distribution over neural responses $r$ ~ $p(r|x)$.
>
> Concretely, one could instantiate an NSC model, generate synthetic input–output data from it (e.g. spike counts or rates for many neurons and stimuli), and treat each “neuron” or condition as a task in our meta-learning setup. With a suitable likelihood (e.g. a point-process/Poisson), one could use our framework in the same way that we do for efficient coding.
>
> In this setting, **the meta-learned GP would be a surrogate, function-space representation of the NSC model**. It’d distill the NSC predictions into a deep kernel with a tractable marginal likelihood. That surrogate could be useful when the kernel is easier to analyze or compare than the original NSC model (e.g. inspecting the kernel, doing kernel PCA, or mixing NSC-informed and generic kernels to quantify “how NSC-like” individual neurons are).
>
> At the same time, NSC theories are already formulated as explicit probabilistic models over neural activity. When one’s primary goal is to test detailed sampling predictions (e.g. stimulus-dependent variability and correlations), approaches that work directly with the NSC likelihood and posterior, such as the recent work of Shrinivasan et al. (2023; discussed in SI A), may be more natural. Our framework is particularly advantageous for normative theories that are not already expressed as full probabilistic models, where a meta-learned GP surrogate adds a probabilistic element that would otherwise be missing.
>
> ## Re: Multiple model instances
> **Yes**, we believe it would be viable, and an interesting research direction. Our method is agnostic to where the meta-training tasks come from: in principle, each instance of a theory (e.g. an efficient-coding model trained on a particular natural image ensemble) simply defines a collection of input–output functions that can be treated as tasks.
>
> Concretely, if we generate, say, 500 “neurons” from one efficient-coding model and 500 from another (trained on a different natural image set or with slightly different architectural choices), we can interleave all of these tasks during meta-training. In that case, the shared feature extractor is encouraged to learn an embedding that captures the structure common across all those instantiations, while the task-specific linear heads and GP hyperparameters account for the differences between particular model instances or neurons.
>
> The resulting kernel would then, in principle, reflect a family of efficient-coding instantiations rather than a single one, and one could compare such families to each other or to alternative theories on real data. In practice, how well this works would depend on how much common structure there actually is across the chosen instances, so getting the meta learning to work well in potentially heterogeneous settings is an interesting open problem
>
>
> ## Thanks again for your constructive assessment, and the thoughtful questions!

---

> > ### Comment · Reviewer_Xmij · 2025-11-27
> > **Thanks for the through response**
> >
> > I thank the authors for aptly addressing my concerns and questions. I would be happy to keep the score where it is. Nice work!

---

### Official Review · Reviewer_YV9C · 2025-11-03

**Soundness:** 4
**Presentation:** 4
**Contribution:** 3
**Rating:** 8
**Confidence:** 4

**Summary:**

This paper introduces a Bayesian meta-learning framework that automatically converts predictions from normative theories (such as efficient coding in vision) into tractable probabilistic models using adaptive deep kernel Gaussian processes. The approach has three stages: (1) generating synthetic data from a normative theory, (2) meta-learning a theory-informed kernel (TIK) on this synthetic data, and (3) adapting this kernel to real biological data. The authors demonstrate their framework on mouse retinal ganglion cell (RGC) responses to natural images, showing improved prediction accuracy over baselines while providing well-calibrated uncertainty estimates and interpretable representations. A key contribution is enabling rigorous Bayesian model comparison to quantify the degree of theory-match in biological data.

The main contributions are: (1) a general framework for automatically constructing probabilistic models from "black-box" normative theories, (2) demonstration that theory-informed priors improve predictive accuracy and uncertainty quantification on real neural data, (3) interpretable representations via prototype images that reveal what task-specific heads learn, and (4) a principled method for validating competing scientific hypotheses using exact marginal likelihoods.

**Strengths:**

- Originality: the paper makes several original contributions. First, it bridges normative modeling and data-driven approaches in a principled Bayesian framework, addressing a longstanding challenge in computational neuroscience. While meta-learning and deep kernel GPs exist independently, their combination for learning theory-informed priors over functions is novel. The adaptive nature of the prior, where the Bayesian Occam effect automatically relaxes theoretical structure as data becomes abundant (Fig. 4b), is a novel finding in line with theory.
- Quality: The experimental validation is thorough and rigorous. The authors provide extensive ablation studies (Fig. S1, Table S1) demonstrating that performance gains specifically arise from meta-learned theory-informed features, not from generic meta-learning or random features. The framework also successfully avoids feature collapse (a known pathology in deep kernel GPs) through careful design choices that are well-documented (S.F.4). The validation on synthetic data with controlled optimality levels (Fig. 5a, showing Pearson correlation of 0.88 with ground truth) nicely demonstrates that the method can accurately infer the degree by which the data matches the theory. I am also pleased that the paper also includes negative results and limitations (the "worst NLPD" neurons in Fig. 4c), which increases credibility.
- Clarity: The paper is generally well-written with clear motivation and strong visual presentation. Figure 1 effectively communicates the three-stage framework, and Figure 2 provides a detailed architectural overview. The mathematical notation is consistent, with the theory-informed kernel clearly specified. The supplementary material is exceptionally comprehensive, providing implementation details, hyperparameters, and additional results that would help reproducibility. I appreciate that the main text is accessible, but maintains technical rigor (with additional information in the supplementals where necessary).
- Significance: this work provides a principled method for validating and comparing normative theories against biological data, a critical need in computational neuroscience. The framework's generality to any normative model where data is available suggests broad impact. Moreover, the ability to perform exact Bayesian model comparison without manual model specification is a useful methodological advancement. The finding that “theory-informed” models excel across all data regimes, and not just low-data settings, challenges conventional wisdom about kernel methods and supports modern perspectives on inductive biases in deep learning. For the machine learning community, this provides a compelling real-world application of deep kernel GPs.

**Weaknesses:**

1. Validation of this method is restricted to a single biological system (86 mouse RGCs from one retina), testing one normative model of the efficient coding principle. It remains unclear how well this method can generalize to other principles or normative models, and how reliant it is on the specific architectural choices (e.g. CNN feature extractor). To that end, I would have liked to see the method applied to other recent models such as Cadena et al. (2019), Bashivan et al. (2019), or even vision transformers, which would help demonstrate the generality of this method. Moreover, the claim that the framework is “broadly applicable” would be substantially strengthened by demonstrating successful application to at least one additional domain from Table S2 (e.g. V1 simple cells, where sparse coding predictions could be tested, or auditory cortex).
2. There is insufficient analysis of when and why the method fails. The paper shows that performance varies substantially across neurons (Fig. 4c, 5b), but provides limited insight into what distinguishes neurons where the method succeeds from those where it fails. The observation that poorly-fit neurons have less interpretable and less consistent prototype images (Fig. 4c-d) is descriptive rather than explanatory. Failures might be explained by (a) neurons that genuinely don't follow efficient coding, (b) limitations in the specific efficient coding model used (Ocko et al. 2018), (c) the meta-train set not covering the relevant part of theory space, or (d) fundamental limitations of the RBF kernel in the learned embedding space. The paper would benefit from a more systematic analysis to explain the failure modes.
3. I have some concerns with the prototype image analysis (Section 4.5, S.H). First, the normalizing procedure for computing $\Delta_{jk}^{(i)}$ (which normalizes by maximum values) is problematic. For instance, a single distant pair of data points can dominate the normalization, which may induce spurious values in the other data pairs. I suggest seeing how the analysis changes if normalization is calculated through relative distance changes, i.e. $\Delta_{jk}^{(i)} = (D_{jk}^{\phi}-D_{jk}^{(i)}) / D_{jk}^{\phi}$, or through Procrustes alignment. Second, the pixel-wise comparison assumes features exist at the same spatial locations across images, which is a strong assumption given that the feature extractor includes convolutional layers that should be translation-invariant. The visualization method cannot capture features that the head is sensitive to at different spatial locations. I suggest a SVD-based analysis of the head transformation, where $W_i = U \Sigma V^{T}$. The right singular vectors indicate the most transformed direction in $\phi$ space, and for each $v_l$ one can compute the gradient $\partial v_{l}^{\top}\phi_{j} / \partial x_{j}$ to show which features in the image most influence that direction.
4. In the section on Bayesian model comparison, high $\beta^{\*}$ could mean either (a) the neuron follows efficient coding principles, or (b) the neuron’s function is well-described by features useful for natural images generally, or (c) both. To distinguish between these, one could test the model on non-natural stimuli such as white noise, gratings, or synthetic stimuli, where natural image statistics do not apply but where efficient coding principles might still hold. Second, it is not clear what $\beta^{*}=0$ means. Is it that the neuron is best described by the RBF null model, or that neither the TIK nor RBF kernel captures it well.
5. The method extracts linearized receptive fields from the EC autoencoder’s bottleneck via LN model fits. Why is the linear readout the right quantity to meta-learn? Moreover, all 490 meta-train tasks come from a single instantiation of the Ocko et al. model with fixed hyperparameters. The paper acknowledges that “varying these parameters would likely yield different efficient coding solutions” (S.E.1). Would meta-training on a family of EC models with varied hyperparameters improve generalization?
6. From section S.G.3, I see that meta-training tasks around 4 hours, and adapting this to all 86 neurons at all data sizes takes 6-12 hours on a single GPU. For researchers wanting to apply this to hundreds or thousands of neurons, or to test variations of their normative theory, these costs could be prohibitive. I think this paper would benefit from an analysis of how runtime scales with meta-train set size, number of neurons, and data dimensions, and a discussion of possible optimizations such as amortized inference or GP approximations.
7. What other ways could you have partitioned the meta-learning process other than a CNN feature extractor and linear readout? Could you consider alternatives, such as task-specific kernels beyond RBF in the head space, meta-learning the kernel type itself, or task-specific transformations? Some discussion of why you made these specific architectural choices would help.

**Questions:**

1. Can you provide a systematic analysis of what distinguishes neurons where the model performs well versus poorly? Specifically, are failure cases correlated with any known biological properties (cell type, receptive field location/size, response reliability)? For the neurons with low $\beta^*$ values (Fig. 5b), is this because they do not follow efficient coding, or is it to do with model limitations? Can you analyze whether failures occur for neurons whose true receptive fields fall outside the range covered by your augmented meta-train set?
2. How would the SVD approach (mentioned above) compare against the current prototype images? Can you validate that the prototype images capture causally-relevant features, for instance by showing that synthetic images emphasizing these prototypes produce predictable changes in model outputs? Finally, for neurons where prototype images look interpretable (Fig. 4c, left), do they match the receptive fields estimated by traditional methods (e.g., spike-triggered average)?
3. Can you test the model comparison approach on non-natural stimuli (white noise, gratings, synthetic images) that do not possess natural image statistics? If $\beta^{\*}$ remains high for neurons with high $\beta^{\*}$ on natural images, this would provide stronger evidence that the theory informed kernel​ captures specific EC principles rather than natural image features. Have you considered alternative null models? For instance, what happens if you compare against the "Task Ablation" kernel (meta-learned on PCA prediction tasks) instead of the RBF? Would this better isolate theory-specific structure? Finally, for biological neurons with $\beta^{\*}$ near 0, can you determine whether this means the RBF null is actually a good fit, or whether both models fail?
4. You meta-trained on one instantiation of the Ocko et al. model with fixed hyperparameters. Have you explored meta-training on a family of EC models with varied hyperparameters? Would this improve generalization and/or allow the framework to discover which EC hyperparameters best match biology?
5. You use fixed task-specific linear heads of a specific dimensionality. How sensitive are results to this dimensionality? How would using non-linear task-specific transformations affect the results? Also, the narrow lengthscale on the kernel seems important. Is there a principled way to set this?

---

> ### Author Response · Authors · 2025-11-20
> **Reply to Reviewer YV9C Part (1/7)**
>
> Dear Reviewer YV9C
>
> Thank you for your positive, and exceptionally thorough assessment of our paper. We truly appreciate your time investment, and have taken care to benefit as much as possible from your feedback.
>
> You have raised a lot of good points, and we apologize for the length of our reply in addressing them:
>
> **To summarize key edits made,**
>
> **i)** We’ve implemented your ideas for improving our prototype image analysis. We agree that your normalization idea is generally more robust, so we have added a discussion about this in the Prototype images section for future readers. However in our use case we found that our results remained largely unchanged.The comparison is now in (SI H.2, line1869, Figure S.7, teal). We also tried the SVD idea, discussed below.
>
>
> **ii)** We’ve run a more systematic analysis to better distinguish the factors that influence the model’s successes and failures on different neurons, which we have added a section about in (SI B.2, line 1080, Figure S.3, teal)
>
>
> **iii)** We’ve enriched our discussion about runtimes and architectural choices in (line 1784, 1462, teal)
>
> # Response Structure
> Due to character limits, we have split our response into 7 comments.
>
> - Comment 1/7 (This comment) addresses Weakness 1.
> - Comment 2/7 addresses Weakness 3 and Question 2.
> - Comment 3/7 addresses Weakness 2 and Question 1.
> - Comment 4/7 addresses Weakness 4 and Question 3
> - Comment 5/7 addresses Weakness 4 and Question 3 (cont.)
> - Comment 6/7 addresses Weakness 5 and Question 4
> - Comment 7/7 addresses Weaknesses 6,7 and Question 5
>
> # Weakness 1: On using other feature extractors, and other datasets
>
> ## Regarding other feature extractors:
> **We agree** it would have been nice to show this, but there is related work making deep kernel meta learning work with different feature extractor architectures (e.g. Chen et al 2023 use an MLP taking inputs from graph neural network features, molecular fingerprints, and hand designed physical chemistry descriptors of molecules), so **we feel it is reasonable to expect the deep kernel meta learning element to transfer to different architectures quite straightforwardly**.
>
> ## Regarding other biological systems, (shared with reviewer raz1.)
> **We agree** with both of you in this case, it would have been better if we demonstrated our method on another system. As you’ve read in SI C, we do argue that success in the retina implies high odds of applicability in other systems, but it would have been better to show that.

---

> > ### Author Response · Authors · 2025-11-20
> > **Reply to Reviewer YV9C Part (2/7), Weakness 3 and Question 2.**
> >
> > # Weakness 3, Question 2: Improvements for Prototype Image method
> >
> > **Actions:**
> >
> > **i)** We have implemented all of your suggestions for improving the computation of the prototype images.
> >
> > **ii)** We have outlined your improvements to the normalization in (SI H.2, line1869), and validated that our current approach is robust to these changes on our dataset in (line 1890, Fig S.7).
> >
> > ## Regarding your normalization idea,
> >
> > We implemented both of your suggested normalizations for the $\Delta$’s: a relative normalization $\frac{D^\phi−D}{D^\phi}​$ and a scalar Procrustes fit $D ^ \phi − \alpha ^*D $ (with $\alpha$ * minimizing $||D^\phi−\alpha D||_{F2}​$). (Since we work purely with pairwise Euclidean distances, which are invariant to rotations and translations, a full Procrustes alignment on coordinates reduces to fitting a global scale factor on the distance matrices). Keeping the rest of the prototype pipeline fixed, we found that for our dataset the **resulting prototype images are extremely similar** (see Fig. S7 line 1890 for side-by-side examples), and re-running the analyses underlying Figs. 4b and 4d did not change any qualitative conclusions. This suggests that either our dataset has few problematic outliers, or that the later centering/averaging steps in the prototype computation already dampen errors from max-scaling. We agree that the alternative normalizations are more robust in general, and **now recommend them explicitly in the appendix (SI-H.2)**.
> >
> > ## Regarding the Gradient feature attribution,
> > We’ve also **implemented the SVD+gradient idea**. We first **validated our implementation** on (i) a simple LN DoG neuron with a hand-designed center–surround RF, and (ii) the final layer of an MLP trained to approximate this LN neuron on natural and white-noise images; in both cases, the SVD+gradient maps recover the DoG RF. Applied to our DKL heads, however, this method **does not produce RF-like structure**: the resulting maps (weighted sum over several u vectors) are diffuse and show relatively uniform sensitivity, even for neurons where learned LN RF’s and our prototypes show clear center–surround patterns. **This is expected**, because in our model the head does not act as a direct predictive readout; it provides metric embedded points to a downstream RBF kernel so downstream predictions depend on changes in pairwise distances specifically. The gradient/SVD method is therefore well-suited for LN-type models, but it does not directly target the geometric quantity that matters in our DKL-GP setup. Our prototypes are designed exactly for that: they summarize how the head reshapes distances between stimuli, which is why they align with LN RFs for well-fit neurons (Fig. S7)
> >
> > ## Re: Assumptions about image overlap, and implications of CNN layers
> > Our prototype method operates in two steps: (i) it measures how the head changes pairwise distances between images in representation space, and (ii) it visualizes these changes by averaging over image pairs in pixel coordinates. The second step does assume that images share a common pixel grid (which they do in our dataset: all stimuli are $36\times 32$ patches on the same coordinate system), and uses soft pixelwise averaging as a simple way to summarize what jointly moved images “look like”. Convolutional layers in our architecture are translation-equivariant rather than fully translation-invariant (we do not use global pooling), so spatial location is still meaningful in the internal representation. **Empirically**, for neurons that are well fit by the model, the **resulting prototypes closely match LN receptive fields** (Fig. S7), suggesting that any residual translation tolerance and our use of pixelwise aggregation do not dominate the visualization
> > ## Re: Do Prototypes match traditional receptive fields
> >
> > **Yes.** Thank you for pointing out that we have not included this in the original manuscript. **We now show in Fig. S.7** the LN receptive fields side by side with the prototype images- demonstrating the match.
> >
> > ## Re: Are prototypes causally responsible for driving predictions
> > **Yes.** Since pairwise distances are precisely *the* property of the data embedding $h_i(\phi(x))$ which structures the output of the downstream RBF kernel, and prototype images by design correspond to the shared pixels that influence pairwise distances between embedded images, we are confident that perturbing an image with features that are prototype-like would strongly impact its distance from the other images in the embedding space, and therefore strongly impact the resulting RBF covariance of the response between that image and the others.

---

> > > ### Author Response · Authors · 2025-11-20
> > > **Reply to Reviewer YV9C Part (3/7), Weakness 2 and Question 1.**
> > >
> > > # Weakness 2 and Question 1: Analysis of Low Accuracy Neurons
> > >
> > > Thank you for these important suggestions.
> > >
> > > **Action:** We have added a new analysis in SI B.2 (Fig. S3, text teal) that directly addresses when and why our method fails on particular neurons and how this relates to the interpretation of \beta (we discuss $\beta$ in the next section).
> > >
> > >
> > > ## Systematic comparison of failure across models
> > > In the absence of repeated presentations of the same stimulus, we interpret cross-model agreement as a proxy for neuron-specific difficulty or “response reliability”).
> > >
> > > We first assessed whether accuracy differences across neurons are specific to TIK or shared across baselines. We computed the Spearman rank-correlation between the accuracy (averaged across seeds, correlation across neurons) of the RBF kernel (and PCA ablation) and our TIK: finding mean Spearman $\rho \approx .82.$ : meaning **neurons that are hard to model for the TIK are also hard for the RBF GP and the PCA deep kernel**. This indicates that the dominant source of performance variation across neurons is shared (we speculate e.g. intrinsic variability or measurement noise), rather than a fundamental limitation specific to our theory-informed kernel.
> > > We have **not yet found any simple dependence beyond this on the basic neuron-level quantities available** in this dataset (e.g., mean firing rate, LN-identified receptive fields), and suspect we cannot go deeper without additional sources of biological metadata (e.g. “true” receptive fields) for these recordings.
> > >
> > > **Regarding your idea (c)** that the meta-train set may not cover the relevant part of theory space, we do think our coverage is sufficient to achieve generalization within this Ocko et al set of theories, as our model transfers very well to unseen instances of the same theory (See, e.g. the closeness of “meta-nlpd” to “meta-nlpd-in” in Fig. S6). We cannot, at the moment, distinguish between (a) and (b), because our framework works with particular instantiations of theories, not the notion of efficient coding in full generality (see, e.g. our response to reviewer Xmij for more).

---

> > > > ### Author Response · Authors · 2025-11-20
> > > > **Reply to Reviewer YV9C Part (4/7), Weakness 4 and Question 3**
> > > >
> > > > # Weakness 4 and Question 3: On $\beta^\*$ and EC vs Generic Natural Image Stats
> > > > While we cannot directly apply our trained model to e.g. white noise stimuli, because the feature extractor was not trained on that input domain and likely would not reliably generalize, we do think we can address the core question of distinguishing natural image statistics from EC.
> > > >
> > > > **Action:** We have **added an additional analysis in SI B.2** (Fig. S3 b) that directly examines how the per-neuron $\beta^*$ relates to predictive performance of different kernels on the biological RGCs, and how much of this effect can be explained by generic natural-image structure.
> > > >
> > > > ## Re: Natural Image Statistics vs EC
> > > >
> > > > To understand what distinguishes neurons where the theory helps most, we stratified cells by their inferred $\beta^\*$ from the TIK–RBF mixture (Fig. 5b). In SI B.2 (Fig. S3 b) we report the mean test correlation of TIK, the PCA-ablation deep kernel, and the RBF GP on the 10 neurons with the lowest and highest $\beta^\*$. For the **10 lowest-$\beta^\*$ neurons, all three models perform similarly poorly** (mean Pearson $r \approx .138, .136, .117$ for TIK, PCA, and RBF), indicating that in these cases neither the EC-informed features nor the alternative deep or RBF kernels provide a strong inductive bias. In particular, in this dataset the lowest $\beta^\*$ correspond to neurons that are difficult to predict for all tested models, rather than to neurons where the RBF null alone is a good standalone description. (On synthetic neurons, where we can observe $\beta^\* \approx 0$, we indeed find that the RBF GP tends to outperform TIK, as expected.)
> > > >
> > > > By contrast, **for the 10 highest-$\beta^\*$ neurons, accuracy improves for all models but in a graded way**: relative to the low-\beta^* group, the RBF GP improves by ~$0.16$ in Pearson r, the PCA ablation by ~$0.24$, and TIK by ~$0.32$. Thus, neurons that our BMC procedure classifies as highly “optimal” are precisely those for which the **EC-informed kernel yields the largest additional predictive gain, with RBF < PCA < TIK**. Since the PCA deep kernel is explicitly meta-trained to capture natural-image structure, this graded pattern is **difficult to explain by generic natural-image features alone** and is consistent with high $\beta^\*$ identifying neurons whose responses are especially well captured by the EC-derived embedding.

---

> > > > > ### Author Response · Authors · 2025-11-20
> > > > > **Reply to Reviewer YV9C Part (5/7), Weakness 4 and Question 3 (cont)**
> > > > >
> > > > > ## Re: Alternative null models (e.g. PCA as null).
> > > > > **We agree** that, in principle, one could define a mixture between TIK and a natural-image deep kernel such as the PCA ablation, and use BMC to compare those. However, doing so would change the semantics of $\beta^\*$ from “preference for an EC-informed kernel over a minimally structured, task-agnostic smoothness prior” to “preference for EC features over this particular representation of natural-image statistics”, which depends strongly on the chosen representation (e.g. PCA vs VAE , domain, etc.). Moreover, because efficient coding is itself a theory about optimal representations of natural-image statistics, **a natural-image deep kernel is not a clean “non-theory” null**- it already captures part of the structure that the EC model aims to explain. Hence such a $\beta^\*$ would be challenging to interpret directly.
> > > > >
> > > > > In this work our goal was to develop a framework that can be applied across domains, so we use an RBF kernel as a simple, principled null and treat the PCA deep kernel as an interesting domain-specific baseline. **Rigorously exploring richer nulls tailored to specific theory classes is a natural next step**, but would require re-calibrating the models and BMC procedures across a number of possible design choices, as well as larger, higher-SNR datasets, which we view as **beyond what can be meaningfully added during rebuttal**.
> > > > >
> > > > >
> > > > > ## Re: The meaning of $\beta^\*$
> > > > > Mathematically, $\beta^\*$  is just the mixing weight that maximizes the marginal likelihood under the kernel $K_\beta = \beta K_{TIK} + (1−\beta) K_{RBF}$; it quantifies the relative contribution of the EC-informed versus RBF kernels in the GP prior that is most supported by the data.
> > > > >
> > > > > On synthetic neurons with known ground-truth optimality, we show that $\beta^\*$ closely tracks a DoG-based measure of EC optimality (Fig. 5a and S.I.1), and that ablation kernels (PCA/random/identity) do not recover optimality under the same BMC protocol. This justifies interpreting $\beta^\*$ as an “optimality score” in that controlled setting. In the biological setting, noise in the dataset adds another dimension to consider when interpreting $\beta^\*$.
> > > > > - **Low $\beta^\*$** indicates that the data suggests a preference for the null. (We have seen this on our synthetic data- the RBF for those \beta values simply has higher accuracy. We have not observed biological neurons with \beta values near zero, which makes sense given that EC is rather well validated for biological neurons)
> > > > > - **High $\beta^\*$** indicates that the data suggest a preference for the TIK. (Seen both on our synthetic data and biological data)
> > > > > - **Intermediate $\beta^\*$** values, like those found at the lower end of the biological $\beta^\*$ range (~0.6), can reflect either genuinely intermediate structure (as in our Synthetic case) or limited discriminative power between kernels under noise (which is plausible for our biological case)
> > > > >
> > > > >
> > > > > For these reasons, in the biological setting we focus our conclusions on patterns across neurons and at the extremes of the $\beta^\*$ spectrum, where the implications are clearer. In particular, it is sensible that many RGCs obtain relatively high $\beta^\*$, which is consistent with independent evidence that efficient coding principles broadly apply in the retina.

---

> > > > > > ### Author Response · Authors · 2025-11-20
> > > > > > **Reply to Reviewer YV9C Part (6/7),  Weakness 5 and Question 4**
> > > > > >
> > > > > > # Weakness 5, Question 4: On the Meta Train set
> > > > > > ## Re: Why Linear RF’s
> > > > > > We chose to focus on linear receptive fields for **two reasons**. **i)** Because they isolate the specific, generally well validated structural predictions of the Efficient Coding (EC) theory (e.g., center-surround shapes)  **ii)** Because, as we discussed in (SI C.1), our deep kernel model intrinsically tries to learn a metric embedding informed by latent variables in which relationships between responses to embedded points become smooth and stationary. For Efficient Coding neurons, these receptive fields roughly correspond to the latent variables in question, making them the natural feature to use.
> > > > > >
> > > > > > ## Re: Would varying hyperparameters improve generalization?
> > > > > > We agree that **training on a diverse family of EC models is desirable for improving generalization, provided that the resulting family of predictions in the meta-train set structurally better reflects the real biological system**. In fact, we implemented a computationally efficient proxy for exactly this. As detailed in SI.E.2, we introduced the scaling augmentation to the optimal RFs in our meta-train set. In Efficient Coding theory, the spatial scale of an optimal RF is largely determined by the assumed input noise level (higher noise -> larger RFs to integrate signal). Therefore, **by scaling the RFs, we effectively meta-trained on a family of solutions corresponding to varying noise hyperparameters**, without the computational cost of training many separate autoencoders. In our case, as seen in the corresponding ablation of the scaling (S.E.2), **this did not yield significant changes**. However, the **space of possible hyperparameters is large, and we did not explore varying them** to improve performance due to computational costs. This is a great direction for future work.
> > > > > > ## Re: Would varying hyperparameters allow the framework to discover which EC hyperparameters best match biology?
> > > > > > **In principle, yes.** Our framework provides the mathematical machinery (via exact Marginal Likelihood) to arbitrate between different theoretical instantiations. While our current study focuses on the more coarse-grained distinction between "Theory" vs. "Null" , distinguishing between fine-grained hyperparameter settings is a logical next step. However, in practice, this requires high statistical power to resolve subtle theoretical differences against biological variability and measurement noise. We view our framework as a necessary step toward this goal, enabling future work to rigorously identify “most biological” hyperparameters as neural recordings increase in scale and SNR.

---

> > > > > > > ### Author Response · Authors · 2025-11-20
> > > > > > > **Reply to Reviewer YV9C Part (7/7), Weaknesses 6,7 and Question 5**
> > > > > > >
> > > > > > > # Weakness 7, Question 5: Engineering Choices
> > > > > > > Thank you for the question.
> > > > > > >
> > > > > > > Most of our choices were derived from standard practice, but we do think the **general design principle** of minimizing the flexibility of the task-adaptive components in a bilevel-optimization setting is **worth pointing out** for future practitioners.
> > > > > > >
> > > > > > > **Action:** We have added (line 1462, Teal) in the SI.
> > > > > > >
> > > > > > > We chose the **dimensionality of the task-specific linear heads** following common practice in deep kernel models: the head dimension is large enough not to over-compress the shared representation (e.g. as discussed by Van Amersfoort et al., 2021), while still providing a compact space for the GP. Within the usual range recommended for deep kernel GPs, we **do not expect this choice to substantially affect performance**.
> > > > > > >
> > > > > > > **Nonlinear task-specific transformations,** assuming they are more flexible than our linear heads, may be detrimental to the success of our meta-learning process. Since we are using bilevel-optimization during meta training, the task-specific components are always first to adapt to the data of a new task-batch. Hence, if they are too powerful, they may obviate the need for the feature extractor to learn a useful representation.
> > > > > > >
> > > > > > > **The narrow lengthscale prior** is a gaussian with a small variance centered at the median lengthscale of projected data points.This technique of initializing the lengthscale of an RBF as the “median” distances between the data points, is **standard in RBF GP’s**.
> > > > > > >
> > > > > > > **Re: Other task-specific kernels,** we did experiment with Matern kernels but found they made hardly any difference. **We deliberately avoided expressive kernels**, like the spectral mixture kernel. This was for **two reasons**. **i)** Kernels like the SMK can be unstable to optimize, and stability of the inner optimization level is important for the bilevel-optimization process. **ii)** An overly-expressive kernel would reduce the pressure on the shared feature extractor to learn useful structure
> > > > > > >
> > > > > > > **Re: CNN feature extractor/linear readout**, the linear readout is important for the above mentioned reasons. However, we chose the CNN as our feature extractor simply because it felt like a natural first choice for image inputs. If our dataset had been structured differently, we would have chosen something else (e.g. graph neural network for molecules, mlp or 1d cnn for audio waveforms)
> > > > > > >
> > > > > > >
> > > > > > >
> > > > > > > # Weakness 6: Scaling the method
> > > > > > > Thank you for pointing this out.
> > > > > > >
> > > > > > > **Action:** We answer your point below, and add a discussion (with refs) to the runtimes section (SI 1784) for future readers.
> > > > > > >
> > > > > > > The reported 6–12h adaptation time corresponds to adapting to all $86$ neurons and all $10$ data sizes in one sweep, with unoptimized code. In practice, the per-neuron, per–data-size cost is modest, and we expect that straightforward engineering (e.g.precomputing CNN features instead of repeatedly applying the base feature extractor, and using lower-precision numerical linear algebra for the GP solves) could significantly reduce the wall-clock time if needed. Our method is also **embarrassingly parallel across tasks**, so each neuron’s adaptation can be run asynchronously depending on compute availability.
> > > > > > >
> > > > > > > In general, **runtime is dominated by two factors**. **(i)** The GP head uses exact RBF inference, which scales as $\mathcal{O}(N^3)$ in the number of data points N per neuron **(ii)** The feature extractor scales roughly linearly in $N$, via standard CNN forwards/backwards. Meta-training time then scales approximately linearly in the number of meta-training tasks and inner-loop steps, and adaptation time scales linearly in the number of neurons and data sizes (again with the cubic term in NN coming only from the GP). The $N^3$ scaling **can be improved by standard tools like sparse/inducing-point GP approximations** (e.g. Titsias et al 2009.)
> > > > > > > So, for $\mathcal{O}(1000)$ data points, with $\mathcal{O}(1000)$ neurons, even our un-optimized code should be able to run within the 6-12 hour range, but speedups are available.
> > > > > > >
> > > > > > >
> > > > > > > # We thank you again for helping us improve the paper with your careful and constructive review. We hope the new analyses and clarifications above address your concerns.

---

### Official Review · Reviewer_raz1 · 2025-11-05

**Soundness:** 3
**Presentation:** 2
**Contribution:** 3
**Rating:** 6
**Confidence:** 3

**Summary:**

The paper proposes a framework for Bayesian meta-learning that turns predictions from a normative theory into a deep-kernel Gaussian Process. The shared feature extractor is meta-trained on synthetic data from an efficient-coding model of the retina, which is then adapted per task to fit real neural activity. The theory-informed kernel enables Bayesian model comparison via a mixture kernel that estimates each neuron’s degree of alignment with the theory. On retinal ganglion cells, the model has higher predictive accuracy and better uncertainty calibration compared to baselines (RBF-GP, CNN, and LN), and in addition, yields interpretable features.

**Strengths:**

- The novelty of this study resides in the development of a Bayesian meta-learing framework that links normative theories to empirical observations.

- The study is technically sound, although I have a few points I would appreciate the authors to address (see weaknesses).

- In general, the manuscript is clearly written.

- The results in this study suggest that the developed approach is a valuable contribution towards integrating normative theories and data-driven modelling.

**Weaknesses:**

- The experiments are limited to a single dataset. It would be desirable to study how the approach generalises to other problems.

- It is unclear how the approach will scale to the very common scenario in neuroscience of the study of neural circuits.

**Questions:**

- Could the authors discuss how the proposed framework might extend to other neural systems beyond the retina, especially in cases where normative theories differ from efficient coding or are less well defined?

- How does the method scale with increasing dataset size or when modelling populations of neurons?

---

> ### Author Response · Authors · 2025-11-20
> **Reply to Reviewer raz1**
>
> Dear Reviewer raz1,
>
> Thank you for taking the time to assess our work. We are grateful for your support and feedback.
>
> Below we address some of your questions/weaknesses:
> # Weaknesses and Questions
> ## Re: Extensions to other systems
> **We definitely agree with the referee on this point**: it would be better if we had also demonstrated our methodology on a system beyond the retina. In practice, we confined our scope to the retina because it is a case where a relatively well validated normative theory, an accessible high-quality dataset, and strong established baselines are all available in a single setting, which allowed us to cleanly isolate the contribution of the framework itself. Still, we believe this framework should work in a broader range of cases.
>
> As we discuss in Sec. S.C of the SI, the modelling assumption imposed by the deep kernel formulation itself is that the data generating process can be made “simple” once inputs are mapped into an appropriate feature space: after passing the stimulus through a learned embedding $h(\phi(x))$, the mapping from $h(\phi(x))$ to the neural response is relatively smooth and approximately stationary, so that a simple kernel (here, an RBF) is a reasonable prior. In other words, we are assuming that each neuron’s response to high-dimensional inputs is effectively governed by some latent structure (in our retina case, receptive-field–like properties), and that $h(ϕ(\cdot))$ can capture that structure in a way that makes the underlying function smooth.
>
> Since there is nothing retina-specific about this assumption, **our success in the retina is consistent with applicability to other domains** where this latent variable story is a reasonable modelling assumption. This is in contrast to, for instance, frameworks that directly constrain a particular part of the model to match a specific system (e.g. the CNN filter constraints of Qiu et al. 2023, SI A- tailored to the retina).
>
> **In Table S.2 (in Sec. S.C) we list other systems** that often are modelled with such assumptions. Here, we think our pipeline could be reused mostly as-is.
>
> **In cases where the theory is not well defined**, as long as there are candidate theories that can generate example input/output predictions, our framework is arguably even more useful – as it can help arbitrate between different ill-defined theories and guide the field toward better-validated models via Bayesian model comparison.
>
>
> So, while we’ve tested our framework on this one system, *we do think that it’s strongly suggestive of broader applicability in principle*.
>
> **Nonetheless, we agree that a demonstration would have been better.**
>
>
> ## Re: Could we do Populations / neural circuits, and bigger datasets
> Yes, there are solutions designed exactly for these settings that can be “plugged in” to the framework as-is.
>
>
> - **Populations / circuits -> Multi Output GP’s**: If the normative theory produces vector-valued (population-level) predictions, the GP can be generalized from $X\rightarrow \mathbb{R}$ to $X\rightarrow \mathbb{R}^d$ using *multi-output* GP’s (e.g. Alvarez et al 2011) on top of the same deep kernel. The meta-learning loop (synthetic tasks, shared $\phi$, task-specific heads) remains unchanged, except each “task” is now a population model: modelling a vector of correlated outputs rather than a scalar.
>
> - **Large datasets -> Sparse/Variational GP’s**: The bottleneck for applying standard GP’s to large datasets is the exact inference, which scales as $\mathcal{O}(N^3)$. To scale to larger datasets, exact GP inference can be replaced by one of the many sparse / variational GP approximations (e.g. inducing-point methods, Titsias et al 2009) that are standard in the deep-kernel GP literature (e.g. Ober et al 2021) and scale as $\approx \mathcal{O}(NM^2)$ with $M<<N$. We’ve added a brief discussion of this to SI G.3.
>
>
> ## Thanks again!

---

> > ### Comment · Reviewer_raz1 · 2025-11-27
> > **Thank you for the thorough responses**
> >
> > I thank the authors for the effort to address the concerns of all the reviewers. I would be happy to raise my score to reflect the overall quality of the paper.

---

### Meta-Review · Area_Chair_pL9K · 2025-12-09

**Summary:**

Dear authors,


The reviewers were, unanimously, positive about this submission. After reading the reviews, your responses, and the paper itself, I am following their recommendation. The paper addresses an important problem to it and provides a clear answer to it, and demonstrate it on one empirical problem.

The reviewers have mixed opinions about whether the paper is well written, with some lauding the clarity of the paper, and others being much more critical. I do think that the paper could have been presented in a clearer framing: Conceptually, what the paper suggests is pretty straightforward: Generate simulations from your 'theoretical' model, use it to pre-train a GP, use that pre-trained GP on real data. The application the authors target is to use 'normative' models from neuroscience to generative the simulations-- however, there is (to my understanding) actually nothing in the approach that requires this and/or uses this, the simulations could come from any model. Therefore, this paper is conceptually related to a lot of the 'sim2real' work (e.g. in robotics) and simulation-based inference which both take a similar approach. I do think that the specific approach, and in particular the problem they are addressing, is new and valuable. However, discussing these relationships, and situating the work in this broader context, would be helpful and important. I ask the authors to take this into account when preparing the final version of their manuscript.

Similarly, one reviewer withdrew from the review process as they did not understand the paper, and the previous AC followed that request. I actually think there is an important signal in the reviewer's comment which should not be discounted-- the paper could, and should, be written in a way that this reviewer could understand it. For example, section 3 is a complete 'jargon-fest' in ML jargon, other sections using Neuro-jargon.  I see no reasons why some of these passages could have been written using plain language that is understandable to both neuroscientists (who are really the target-audience here) and machine learning scientists.

Please do take the input of the reviewers into account when preparing your final submission. I hope you find this feedback useful, and congratulations on this paper!

**Reviewer Concerns:**

Please seem my summary meta review above.

**Reviewer Scores:**

I am unwilling to follow this request. The ACs are being asked to perform an unreasonable amount of extra work this year. In particular, I understand the concern about potential collusion. However,  I do think that asking ACs to now read a whole new set of papers, reviews and (extremely long...) rebuttals and to try to condense them into decisions without a chance for discussions is both asking a lot from us, and will, inevitably, result in (on average) poorer decisions for _everyone_.

I do not see any value in spending even more time to try to 'guess' how each reviewer would have changed their mind or not. I am just trying to make appropriate decisions on the paper and focus on the science. The authors can see the reviewer inputs and should take it into account.

---

### Decision · Program_Chairs · 2026-01-26

Accept (Poster)